# Photography Perspective Composition: Towards Aesthetic Perspective Recommendation

**Lujian Yao**[*] **Siming Zheng**[†] **Xinbin Yuan** **Zhuoxuan Cai** **Pu Wu**
**Jinwei Chen** **Bo Li** **Peng-Tao Jiang**[†][✉]
vivo Mobile Communication Co., Ltd
lujianyao@mail.ecust.edu.cn, pt.jiang@vivo.com
Project page: https://vivocameraresearch.github.io/ppc

## Abstract

Traditional photography composition approaches are dominated by *2D* cropping-based methods. However, these methods fall short when scenes contain poorly arranged subjects. Professional photographers often employ perspective adjustment as a form of *3D recomposition*, modifying the projected 2D relationships between subjects while maintaining their actual spatial positions to achieve better compositional balance. Inspired by this artistic practice, we propose *photography perspective composition* (PPC), extending beyond traditional cropping-based methods. However, implementing the PPC faces significant challenges: the scarcity of perspective transformation datasets and undefined assessment criteria for perspective quality. To address these challenges, we present three key contributions: (1) An automated framework for building PPC datasets through expert photographs. (2) A video generation approach that demonstrates the transformation process from less favorable to aesthetically enhanced perspectives. (3) A perspective quality assessment (PQA) model constructed based on human performance. Our approach is concise and requires no additional prompt instructions or camera trajectories, helping and guiding ordinary users to enhance their composition skills.

## 1 Introduction

Professional photography demands expertise in multiple aspects, with photographic composition being one of the most crucial. Photographic composition refers to the arrangement of visual elements according to aesthetic principles. It requires photographers to harmoniously integrate multiple elements like people, urban, and natural features. Master photographers, such as those in Magnum Photos, require professional knowledge and extensive training, making quality photography expensive and challenging for ordinary people. This raises the question: Can we help ordinary people achieve professional-level composition?

Traditional photography composition approaches are primarily based on cropping. Numerous approaches have been developed for image cropping, including saliency-based methods [36], learning-based techniques [5, 8, 11, 20, 26, 27, 42, 51], and reinforcement learning strategies [19]. However, crop-based methods are inherently limited, as they only allow for 2D recomposition within the image plane. As shown in Fig. 1a, traditional crop-based methods primarily focus on learning a crop template. However, when the scene itself is chaotic and lacks good compositional structure, cropping alone rarely produces satisfactory results.

---

[*]Intern at vivo Mobile Communication Co., Ltd.
[†]Project lead.
[✉]Corresponding author.

39th Conference on Neural Information Processing Systems (NeurIPS 2025).

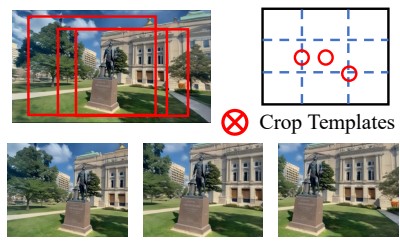

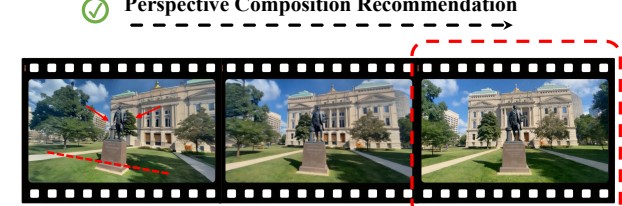

(a) Crop-based Photo Composition | (b) Our Perspective-based Photo Composition

Figure 1: The motivation for the proposed photography perspective composition (PPC). Traditional crop-based methods (a) focus on learning crop templates for better composition. However, when scenes contain chaotic arrangements of subjects, cropping alone rarely yields satisfactory results. Perspective transformation (b) addresses these challenges by adjusting spatial relationships between subjects (e.g., person and tree, red arrow) and scene orientation.

In real-world scenarios, photographers address these limitations of 2D cropping by actively adjusting their perspective and positions to achieve improved spatial relationships between subjects. Through perspective adjustments, photographers can create sophisticated compositions by systematically arranging subjects within the frame, manipulating spatial relationships to create dynamic and engaging images. Fig. 1b illustrates how perspective transformation can address compositional challenges.

Inspired by this artistic practice, we introduce *photography perspective composition* (PPC) as a new paradigm for photography composition. However, implementing PPC presents three main challenges: ❶ Data acquisition is particularly challenging as currently available datasets are limited to planar crop data and lack perspective transformation information. ❷ The implementation of perspective recommendation requires careful design considerations, as compositional aesthetics often follow partial ordering rather than total ordering relationships. ❸ The aesthetic evaluation of different perspectives requires new metrics and evaluation methods.

To address these challenges, we propose three key solutions. First, we develop a novel method for constructing aesthetic perspective transformation datasets, with an *automated data generation pipeline* (for ❶). Second, we implement a perspective transformation video generation approach instead of single-image recommendations. This enables before-and-after compositional comparisons while providing users with intuitive visual guidance (for ❷). Finally, we construct a comprehensive perspective quality assessment (PQA) model that evaluates perspective transformation quality through three critical dimensions: visual quality, motion quality, and composition aesthetic (for ❸).

Our main contributions can be summarized as follows:

- To the best of our knowledge, we are the first to introduce *photography perspective composition*, moving beyond traditional cropping methods. We hope our work can inspire more researchers to explore and advance perspective-based composition techniques in computational photography.

- We develop a concise PPC, without requiring additional prompt instructions or camera motion trajectories, which can help ordinary users improve their photo composition skills.

- We present an automated framework for constructing aesthetic perspective transformation datasets, which leverages large-scale expert photograph collections to learn generalizable principles of aesthetic composition.

- A perspective quality assessment (PQA) model is constructed based on human performance to evaluate the quality of perspective transformation through three aspects: visual quality, motion quality, and compositional aesthetics.

## 2 Related Work

This section reviews two key research areas related to our work: (1) Photography composition, focusing on various image cropping techniques and their data-driven approaches (Sec. 2.1), and (2)

the evaluation with human performance, particularly the recent adoption of vision-language models (VLM) for quality assessment (Sec. 2.2).

## 2.1 Photography Composition

Prior photography composition methods primarily rely on image cropping. There are three types. (1) free-form cropping. Numerous techniques have been explored to tackle this problem, including saliency maps [36], learning-based methods [5, 8, 11, 20, 26, 27, 36, 42, 51], and reinforcement learning [19]. (2) Subject-aware image cropping [12, 47], where an additional subject mask is provided to indicate the subject of interest. (3) Ratio-aware cropping [4], where the crops are expected to adhere to a specified aspect ratio. Several datasets have been established, including GAICD [51], CPC [42], FCDB [3], and SACD [47]. Recent advances in diffusion models [32, 54] have further expanded the possibilities, with works leveraging Stable Diffusion for synthetic data generation [33, 35] and developments in outpainting [12, 34, 49]. However, cropping methods share a fundamental limitation as they operate only in 2D space, making them insufficient when the spatial arrangement of the scene is less favorable.

## 2.2 Evaluation with Human Performance

Evaluation models are essential for aligning generative models with human preferences. Traditional metrics like FID [10] and CLIP scores [29] have been widely used [14, 15, 25]. To improve evaluation accuracy, recent works [7, 16, 22, 43, 46, 53] have evolved from simple metrics to learning-based approaches that leverage human preference datasets to train CLIP-based models. Recently, researchers have begun exploring vision-language models (VLMs) [1, 40] as a more powerful framework for reward modeling. These VLM-based approaches have shown success in both evaluation [9, 43, 44] and optimization [18, 28, 38, 46], utilizing methods like point-wise regression [9, 45], pair-wise comparison with Bradley-Terry loss [2], and instruction tuning [21, 23, 41] to leverage reasoning capabilities for VLMs. However, VLMs need substantial data for effective training, yet expert compositional data is scarce and costly to obtain, making it challenging to train VLMs to capture sophisticated aesthetic principles using only limited expert annotations.

## 3 Methodology

### 3.1 Overview

Unlike previous photography composition methods that primarily rely on cropping, we propose a novel approach that recommends photography composition through *perspective transformation*. First, we describe how to construct the dataset and implement an automated pipeline (Sec. 3.2). Then, we present the core implementation methods of our PPC, and incorporate RLHF to make the generated videos aligned with human performance (Sec. 3.3). Finally, we introduce the implementation of the perspective quality assessment (PQA) Model (Sec. 3.4).

### 3.2 Automated Construction of PPC Dataset

[Intuition.] Currently, no dedicated dataset exists for PPC setting. To promote this area, we propose a novel approach for constructing PPC dataset in an automated way. The main challenge lies in collecting real-world camera movement sequences that transition from *less favorable* to *well-composed* perspective. The closest existing data comes from photographers sharing point-of-view (POV) recordings of their shooting process on streaming platforms. However, these videos are typically captured from secondary angles using GoPro, which differs from the main camera perspective. We observe that expertly composed photographs are readily available, and recent advances in single-image scene reconstruction have shown remarkable results [50]. This inspired us to explore an alternative approach: *generating camera movement sequences that transition from well-composed to less favorable perspective based on existing expertly composed photographs*. By *reversing* these sequences, we can obtain the desired data.

[Detail.] (1) Data Source. We select multiple professional photography datasets, including datasets used in existing composition studies such as GAICD [51], SACD [47], FLMS [6], and FCDB [3]. Furthermore, to expand our data volume, we incorporate Unsplash (https://unsplash.com),

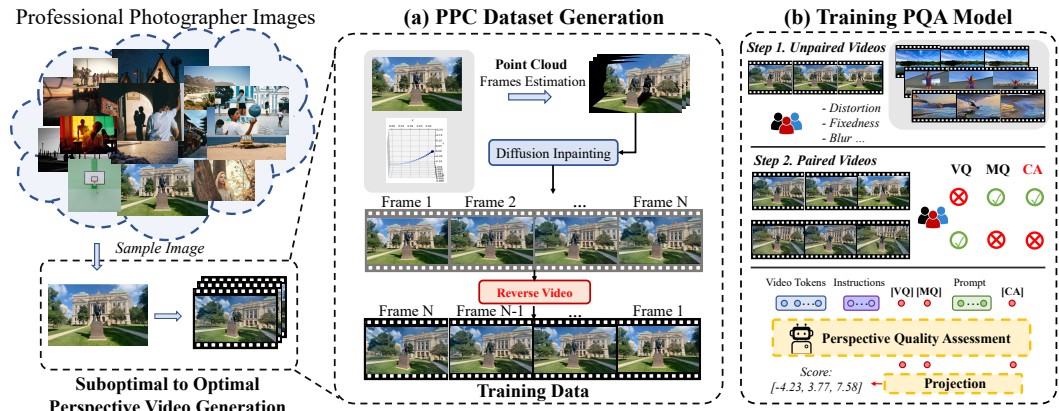

Figure 2: Architecture illustration of PPC dataset generation and the training perspective quality assessment (PQA) model.

Table 1: Grade assessment in data filtering.

| Grade | Level | Raw Score Range | Standardized Score (Range) | Quality Assessment |
|-------|-------|-----------------|----------------------------|--------------------|
| A | Excellent | $\geq 5.0$ | 95 (90-100) | Superior quality |
| B | Good | 0.0 to 4.9 | 85 (80-89) | Above-average quality |
| C | Satisfactory | -5.0 to -0.1 | 75 (70-79) | Acceptable quality |
| D | Marginal | -15.0 to -5.1 | 65 (60-69) | Below-average quality |
| E | Unsatisfactory | $< -15.0$ | 50 ($< 60$) | Inadequate quality |

currently the largest open-source professional photography dataset. **(2) Perspective Transformation Generation.** As shown in Fig. 2, we adopt a 3D reconstruction approach. Our 3D reconstruction methodology mainly builds upon the ViewCrafter [50]. The inputs consist of a well-composed image and a specified camera motion trajectory. Note that this trajectory can be random (refer to **[Discussion]**). By following this trajectory, we can generate a video sequence transitioning from the well-composed to less favorable perspective. Then, by *reversing* this video sequence, we obtain our desired training data. **(3) Data Filtering.** Given the limited performance of current reconstruction models, the generated video data needs to filter out artifacts including distortion, fixedness, and blur effects, as depicted in Fig. 2. However, manual filtering for such a large dataset is impractical. Our tests showed that a single person can only filter about 3K videos per day, making it difficult to process large-scale samples. With the rapid advancement of vision language models (VLMs) in scene understanding and automated evaluation [24, 28, 44], we develop a perspective quality assessment (PQA) model to filter the generated data. For specific details about the PQA construction, please refer to Sec. 3.4. Our PQA evaluates generated data across three dimensions: visual quality (VQ), motion quality (MQ), and composition aesthetic (CA). These individual scores are aggregated into a final score, and samples exceeding a threshold are selected as our training data. We implement a comprehensive grading assessment that converts model-generated scores into standardized grade scores. It employs a five-tier grading scale (A to E) with corresponding numerical ranges, as shown in Tab. 1.

**[Discussion.]** *How to handle cases where the initial perspective is less favorable?* In our pipeline, we initially treat the original image as the well-composed perspective. This assumption is later refined through human preference learning during the PQA filtering (Sec. 3.4) and RLHF (Sec. 3.3) stages, acknowledging that the original perspective may not always be the aesthetically pleasing.

## 3.3 Photography Perspective Composition (PPC)

**[Intuition.]** Previous image composition works primarily focused on cropping approaches. In contrast, we propose a video-based approach: given a less favorable perspective, we generate a camera movement sequence that gradually transitions to an aesthetically enhanced perspective of the scene. This video-based approach is motivated by two key observations: First, image composition represents

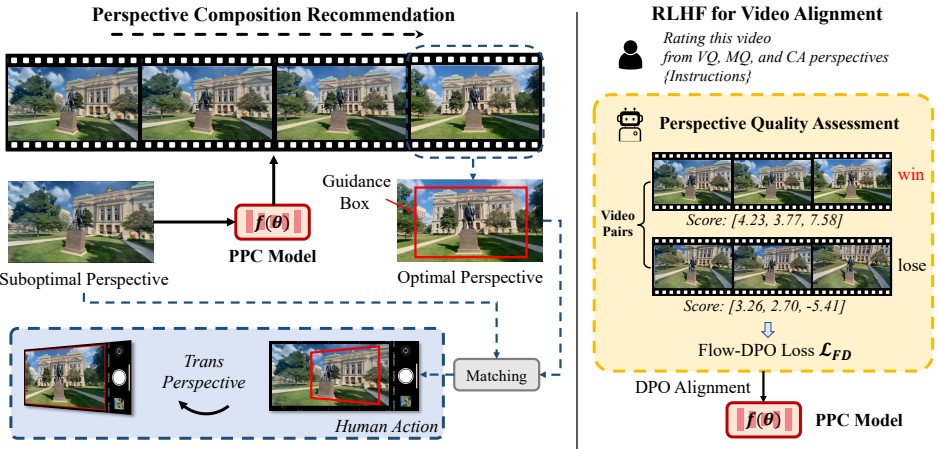

Figure 3: The pipeline of proposed photography perspective composition (PPC).

a partial order relationship where quality assessment often relies on comparing different views of the same scene rather than isolated evaluation. Second, the transition process naturally demonstrates compositional improvements through before-and-after comparisons, making it particularly effective for both visual demonstration and educational purposes.

**[Detail.] (1) Base Pipeline.** As shown in Fig. 3, our pipeline takes a less favorable perspective as input and generates a transformation video from the less favorable to aesthetically enhanced perspective. This process can be modeled as an image-to-video (I2V) task. I2V has seen remarkable progress, with both commercial solutions like OpenAI's Sora (https://openai.com/sora), Runway Gen-3 (https://runwayml.com), and Pika 1.5 (https://pika.art), and open-source models like Hunyuan [17], CogVideo [48], and WAN [39]. Our pipeline leverages these open-source models to generate perspective transformation videos. We utilize the last frame of the video as our final aesthetically enhanced perspective and design a method to guide human actions. First, we draw a guidance box (the red bbox in Fig. 3) on the enhanced perspective. Then, based on this box, along with the initial and final perspectives, we transform this box onto the original image using feature matching, creating a distorted box. As the user moves, this box gradually changes shape, approaching a rectangle when reaching the aesthetically enhanced perspective. To simplify the process and accelerate computation, we only use the traditional homography transformation.

**(2) RLHF for Quality Enhancement.** We observed that some generated perspective transformation videos, while deviating from the GT direction, still maintain high aesthetic quality. This suggests that strict GT adherence may not always yield the most aesthetically pleasing results. Therefore, we propose incorporating direct preference optimization (DPO) to align the model with human preferences. This approach encourages the exploration of aesthetically pleasing trajectories that may differ from GT, avoiding the limitation where GT-based optimization could discourage potentially superior compositional alternatives.

Our RLHF implementation primarily draws from Diffusion-DPO [30] and VideoAlign [24]. Consider a fixed dataset $\mathcal{D} = \{s, v_h, v_l\}$, where each sample consists of a prompt $s$ and two videos, $v_h$ (higher-quality video) and $v_l$ (lower-quality video), generated by a reference model $p_{\text{ref}}$, with human annotations indicating that $v_h$ is preferred over $v_l$ (i.e., $v_h \succ v_l$). The goal of RLHF is to learn a conditional distribution $p_\theta(v \mid s)$ that maximizes a reward model $r(v, s)$ (i.e., PQA model proposed in Sec. 3.4), while controlling the regularization term (KL-divergence) from the reference model $p_{\text{ref}}$ via a coefficient $\beta$:

$$\max_{p_\theta} \mathbb{E}_{s \sim \mathcal{D}_c, v \sim p_\theta(v|s)} \left[ r(v, s) \right] - \beta \, \mathbb{D}_{\text{KL}} \left[ p_\theta(v \mid s) \, \| \, p_{\text{ref}}(v \mid s) \right]. \tag{1}$$

In Rectified Flow, we adopt videoalign [24] that relate the noise vector $\xi^*$ to a velocity field $\nu^*$, where $\nu^*$ represents the velocity field of either the higher-quality video ($\nu^h$) or lower-quality video ($\nu^l$). Specifically, it can be proved that $\|\xi^* - \xi_{\text{pred}}(\nu_t^*, t)\|^2 = (1-t)^2 \|\nu^* - \nu_{\text{pred}}(\nu_t^*, t)\|^2$, where $\xi_{\text{pred}}$ and $\nu_{\text{pred}}$ refer to predictions either from the model $p_\theta$ or the reference model $p_{\text{ref}}$. Based on this

Table 2: Comparison of I2V models in generating perspective transformation videos.

| Method | Perspective Accuracy | | | | | Human Performance Score | | |
|---|---|---|---|---|---|---|---|---|
| | CMM ↑ | FVD ↓ | PSNR ↑ | SSIM ↑ | LPIPS ↓ | VQ ↑ | MQ ↑ | CA ↑ |
| CogvideoX 1.5 5B [48] | 0.5501 | 303 | 8.2380 | 0.2611 | 0.7969 | 0.7073 | 0.7311 | **0.7196** |
| Hunyuan I2V [17] | 0.4928 | **264** | **9.4017** | **0.3537** | 0.7915 | **0.7216** | **0.7496** | 0.7070 |
| Wan2.1 14B [39] | **0.5989** | 345 | 9.3668 | 0.3265 | **0.7808** | 0.7195 | 0.7454 | 0.7072 |

| Method | Video Quality | | | | | | | |
|---|---|---|---|---|---|---|---|---|
| | I2V Subject | I2V Background | Subject Consistency | Background Consistency | Motion Smoothness | Dynamic Degree | Aesthetic Quality | Imaging Quality |
| CogvideoX 1.5 5B [48] | 0.9632 | 0.9639 | 0.9545 | 0.9502 | 0.9906 | 0.1347 | 0.5314 | 0.5667 |
| Hunyuan I2V [17] | **0.9866** | **0.9878** | **0.9582** | **0.9663** | **0.9927** | 0.7851 | **0.5583** | 0.5893 |
| Wan2.1 14B [39] | 0.9618 | 0.9694 | 0.9470 | 0.9435 | 0.9917 | **0.8883** | 0.5464 | **0.6299** |

relationship, we obtain the Flow-DPO loss $\mathcal{L}_{\text{FD}}(\theta)$:

$$-\mathbb{E}\left[\log\sigma\left(-\frac{\beta_t}{2}\left(\left(\|\nu^h-\nu_\theta(\nu_t^h,t)\|^2-\|\nu^h-\nu_{\text{ref}}(\nu_t^h,t)\|^2\right)-\left(\|\nu^l-\nu_\theta(\nu_t^l,t)\|^2-\|\nu^l-\nu_{\text{ref}}(\nu_t^l,t)\|^2\right)\right)\right)\right],$$
(2)

where $\beta_t=\beta\left(1-t\right)^2$ and the expectation is taken over samples $\{v_h,v_l\}\sim\mathcal{D}$ and the schedule $t$.

### 3.4 Perspective Quality Assessment (PQA) Model

**[Intuition.]** PQA serves dual purposes: filtering generated training data to enable automated dataset construction, and providing win-lose pairs for RLHF in PPC to align with human preferences. Due to the limitations of 3D reconstruction models, some scenes suffer from inaccurate point cloud reconstruction or inpainting distortions, leading to distortion, fixedness, and blur. We propose a *two-stage* training strategy for the PQA model that addresses the dual challenges of data volume and expertise requirements. Given that fine-tuning VLMs demands substantial training data, the first *unpair-wise* stage leverages large-scale, efficiently collected data to meet this requirement, as basic quality assessment does not demand expert knowledge. The subsequent *pair-wise* stage employs expert-annotated compositional data to refine the aesthetic capabilities of the model.

**[Detail.]** **(1) Dateset Setting.** *Stage ①: Unpaired Videos.* This stage focuses on distinguishing video quality levels. We collected approximately 5K perspective transformation videos generated by 3D reconstruction models, with expert annotators identifying roughly 1.5K high-quality and 3.5K low-quality samples. To expand the dataset, we randomly paired each high-quality video with 10 low-quality ones, creating a 15K unpaired dataset. *Stage ②: Paired Videos.* This stage focuses on composition aesthetic recognition through paired comparison learning. For each initial perspective, we generate three video clips using separately trained CogVideoX 1.5, WAN 2.1, and the original GT data. These clips are paired with each other, where "paired" indicates videos sharing identical input views. Expert annotators evaluate these pairs across three dimensions: visual quality (VQ), motion quality (MQ), and composition aesthetic (CA). For each dimension, annotators choose between options (*A wins/Ties/B wins*). Notably, the CA metric assesses the compositional improvement throughout the video transformation rather than static frame quality. Detailed annotation guidelines are provided in Appendix A.

**(2) Model Setting.** Our model primarily follows the architecture of VideoAlign [24]. We employ Qwen 2-VL as our base model, utilizing the Bradley-Terry model with ties loss (BTT) [31], which extends the traditional Bradley-Terry framework[2]. To better handle multi-dimensional evaluation, we separate special tokens for context-agnostic (visual quality, motion quality) and composition-aware (composition aesthetic) attributes, leveraging the causal attention mechanism to achieve effective feature decoupling. The model predicts rewards for each dimension through a shared linear projection head applied to the corresponding token representations from the final layer.

## 4 Experiments

In this section, we evaluate our approach through two main components: photography perspective composition (PPC) and perspective quality assessment (PQA).

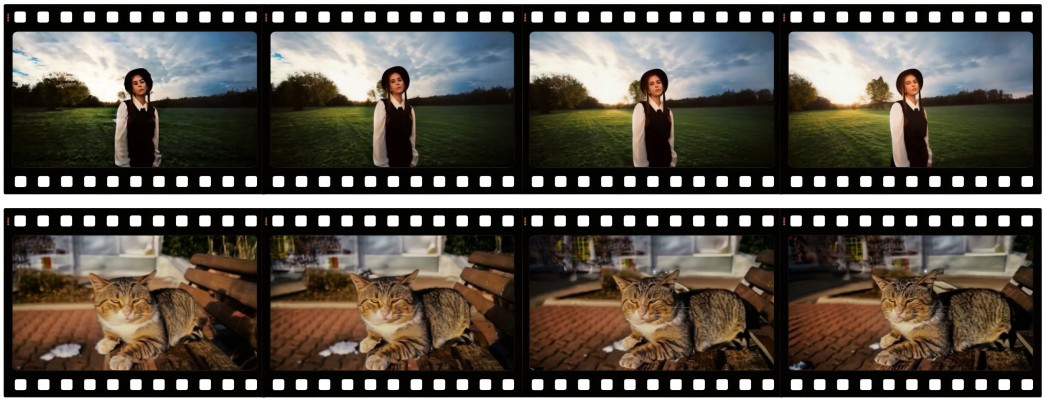

Figure 5: PPC performance in single-subject scenarios.

## 4.1 Investigation for Photography Perspective Composition (PPC)

**Implementation.** To comprehensively validate our approach, we experimented with three state-of-the-art video generation models: CogVideoX 1.5 [48], HunYuan [17], and Wan2.1 [39]. The training parameters follow the settings from the original repository.

**Metric Design.** Since there is no prior work on photography composition using perspective transformation, we need to define metrics. We evaluate the generated videos from three parts: perspective accuracy, video quality, and human performance score. For *video quality* assessment, we primarily adopt the evaluation metrics from VBench2.0 [15] I2V benchmarks, which include I2V subject, I2V background subject consistency, background consistency, motion smoothness, dynamic degree, aesthetic quality, and image quality. For *perspective accuracy* evaluation, we employed both video distance and image similarity metrics. The video distance was measured using camera motion matching (CMM)

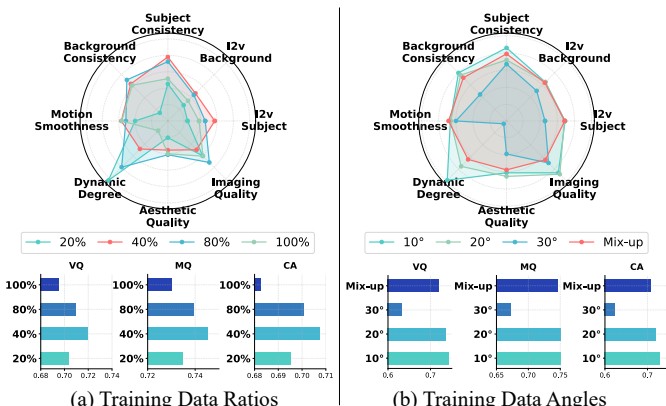

Figure 4: Quality and human performance results for PPC.

and Fréchet Video Distance (FVD) [37] , while image similarity was assessed using CLIP-F [29], PSNR, SSIM, and LPIPS. Notably, we modified the camera motion section of the original vbench [15], changing it to camera motion matching. As we found that the camera motion detection model in vbench did not work well, especially for small angle changes. Instead, we now pass both the prediction and ground truth through the camera motion detection and then calculate the accuracy by matching these two detection results." As for the *human performance score*, its purpose is to simulate human preferences in scoring. This is conducted through PQA, which we established in Sec. 3.4, and includes three components: visual quality (VQ), motion quality (MQ), and composition aesthetic (CA).

**Main Results.** *[Quantitative Results]* As shown in Tab. 2, we demonstrate the performance of three I2V models in generating PPC videos. *[Qualitative Results]* We demonstrate the versatility of our approach across three representative scenarios. The first scenario addresses *single subjects* (e.g., human figures and animals), as shown in Fig. 5, where our PPC model enhances compositional harmony by seamlessly integrating subjects with their surroundings. In the second scenario (Fig. 7), we tackle *multi-subject* scenes, demonstrating how PPC achieves balanced spatial arrangements

Table 3: Quantitative Result for PQA (Tab. (a) and Tab.(b)) and PPC (Tab. (c) and Tab. (d)).

(a) The number of pairs.

| | Accuracy | | |
| --- | --- | --- | --- |
| # Pairs | VQ | MQ | CA |
| 1 | 0.6515 | 0.5957 | 0.5881 |
| 5 | 0.7772 | 0.7812 | 0.7894 |
| 10 | 0.8019 | 0.8085 | 0.8102 |
| 100 | 0.8008 | 0.8063 | 0.8103 |

(b) Different steps.

| | Accuracy | | |
| --- | --- | --- | --- |
| # Steps | VQ | MQ | CA |
| Reg Single | 0.4874 | 0.4986 | 0.4761 |
| Reg Two | 0.7807 | 0.7802 | 0.7935 |
| BTT Single | 0.5509 | 0.5117 | 0.4913 |
| BTT Two | 0.8019 | 0.8085 | 0.8102 |

(c) The data ratios.

| Ratio | CMM ↑ | FVD ↓ |
| --- | --- | --- |
| 20% | 0.5014 | 460 |
| 40% | 0.5989 | 345 |
| 80% | 0.5244 | 362 |
| 100% | 0.5673 | 359 |

(d) The data angles.

| | CMM ↑ | FVD ↓ |
| --- | --- | --- |
| 10° | 0.4413 | 397 |
| 20° | 0.5587 | 337 |
| 30° | 0.3983 | 444 |
| Mix-up | 0.5989 | 345 |

to elevate overall visual aesthetics. The third scenario explores *landscape* photography (Fig. 8), addressing two common challenges faced by novice photographers: balance and horizontal alignment. Our model effectively optimizes these scenes, particularly enhancing symmetrical compositions. Beyond these scenarios, we discovered the applicability of PPC to UAV photography. As illustrated in Fig. 9, our model successfully identifies aesthetically enhanced views from drone-like perspectives, generating camera movements that adhere to compositional principles while maintaining aesthetic appeal.

**PPC Angles.** To maintain high consistency in our recommendation system, we limit perspective transformations to short angles, ensuring our suggestions are reliably based on the actual visible content in the current scene. We also investigated the impact of PPC angles on performance using three distinct rotation degrees (10°, 20°, and 30°) and a balanced mixed dataset incorporating all angles. As shown in Tab. 3d and Fig. 4b, we observe that while performance remains stable at 10°, both quality and accuracy metrics deteriorate significantly when the dataset rotation angles reach 30°. We attribute this degradation to the substantial visual disparity between the original and transformed views at larger angles, which makes it challenging for the model to learn generalized aesthetic perspectives.

**The Consistency of PPC.** Fig. 6b demonstrates our model's consistency in PPC. When presented with different less favorable views of the same scene, our model generates consistent aesthetically enhanced perspectives, maintaining coherence across different inputs.

**The Effection of RLHF in Video Quality Enhancement.** We evaluate the performance after incorporating RLHF (Sec. 3.3). Fig. 6a and Tab. 4 demonstrate the effectiveness of incorporating RLHF. The results show that RLHF leads to more stable subject generation.

Original Perspective    w/o RLHF    with RLHF

(a) Qualitative comparison of RLHF

(b) PPC maintains perspective consistency

Figure 6: Qualitative comparison of RLHF and the perspective consistency of PPC.

## 4.2 Investigation
## for Perspective Quality Assessment (PQA)

**Implementation and Evaluation Metric.** We utilize Qwen2-VL-2B [40] as the backbone for PQA and train it with BTT loss. To fine-tune the model, LoRA [13] is applied to update all linear layers in the language model, while the vision encoder's parameters are fully optimized. The training process is conducted with a batch of 32 and a learning rate of $2 \times 10^{-6}$, with the model trained over two epochs. This setup requires approximately 50 NVIDIA H20 GPU hours. Several observations were made during training. We sample videos at 1 fps, with a resolution of $448 \times 448$ pixels during the training process. Following previous works [24], we adopt accuracy as the metric for each dimension.

**Main Results.** *[Quantitative Result]* To investigate the impact of training data volume in *stage 1*, we conducted experiments with varying numbers of video pairs, as detailed in Tab 3a. Our results reveal that while performance generally improves with increasing sample size, it plateaus at approximately 100 samples, suggesting that this quantity provides sufficient diversity for model performance. *[Qualitative Results]* Fig. 10 demonstrates the basic effects of PQA. As shown in

Table 4: The effect of RLHF in PPC.

| | CMM | FVD | VQ | MQ | CA |
| --- | --- | --- | --- | --- | --- |
| w/o RLHF | 0.4928 | 264.7672 | 0.7216 | 0.7496 | 0.7070 |
| with RLHF | 0.5014 | 270.2212 | 0.7477 | 0.7774 | 0.7342 |

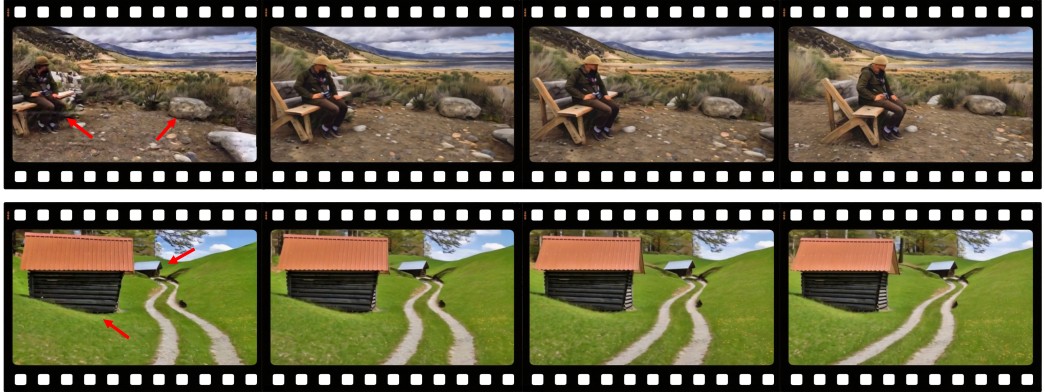

Figure 7: PPC performance in multiple-subject scenarios.

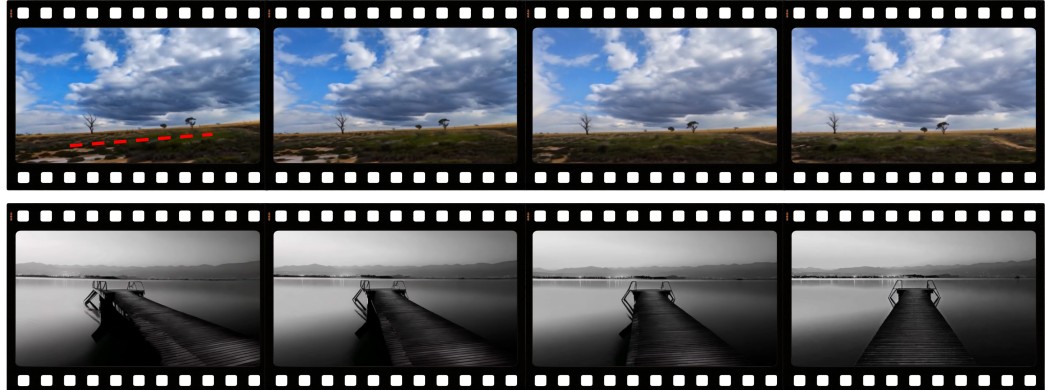

Figure 8: PPC performance in wide landscape views and asymmetric scenarios.

Fig. 4a, while there is a notable improvement in composition, the video quality remains low. Fig. 4b exhibits acceptable levels of both VQ and MQ, but shows minimal compositional differences, resulting in a lower CA score. Notably, PQA assigns lower ratings to cases where the perspective remains static or too intense, as illustrated in the last two examples.

**Effect of Two Steps.** As shown in Tab.3b, we evaluated the effectiveness of the two-step approach under both regression and BTT loss [2, 24] functions. The single-step approach, lacking sufficient data to enhance baseline performance, demonstrates significantly lower overall performance compared to the two-step way.

## 5   Conclusion and Limitation

**Conclusion.** In this work, we addressed the limitations of previous photography composition methods by introducing photography perspective composition (PPC), a novel paradigm that extends beyond 2D cropping to achieve 3D recomposition. Our approach is inspired by real-world street photography practices where photographers use perspective adjustment to establish better relative relationships between subjects. To overcome the challenges of implementing PPC, particularly the lack of suitable datasets and unclear assessment criteria, we made three significant contributions. We developed a framework for automatically constructing a PPC dataset from expert photographs, created a system for generating perspective transformation videos that guide users from less favorable to aesthetically enhanced views, and introduced the perspective quality assessment (PQA) model that evaluates both video quality and compositional aesthetics. We hope this work opens up new possibilities in computational photography and inspires further research in perspective-aware composition.

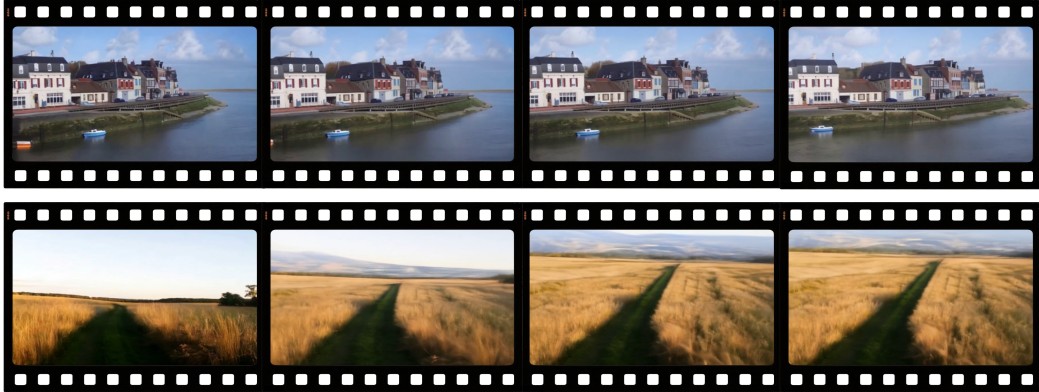

Figure 9: PPC performance in UAV-like scenarios.

**VQ:** -0.009; **MQ:** 5.875; **CA:** 3.578   **VQ:** 5.531; **MQ:** 7.563; **CA:** -1.016

**VQ:** -7.281; **MQ:** -7.500; **CA:** -9.063   **VQ:** -6.656; **MQ:** -5.969; **CA:** -6.750

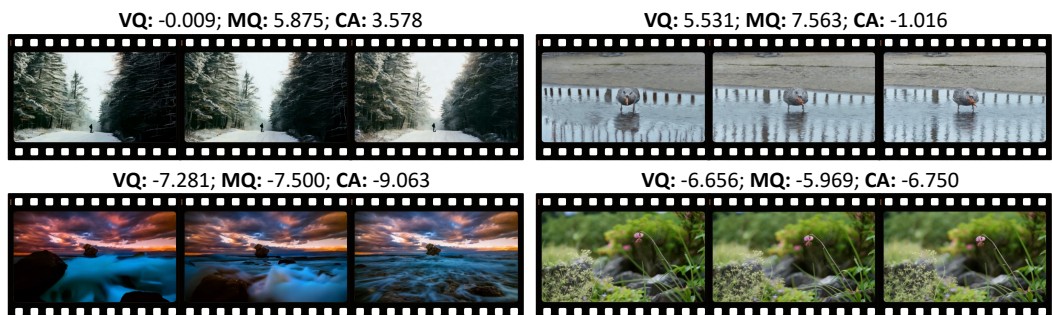

Figure 10: Quantitative Results of PQA.

**Limitation and Future Work.**

**(1) Video Duration.** Current PPC is constrained by the limitations of existing video models, particularly in terms of duration. In the latest phase, we have also discovered AR-based video generation models that can generate infinite streaming videos. We believe this work can provide broader insights and directions for PPC.

**(2) Video Quality.** Since our training data is generated by 3D reconstruction models, the performance of PPC is inherently limited by current reconstruction capabilities. A crucial direction for future improvement lies in exploring superior methods for generating perspective transformation videos, such as utilizing the *Unreal Engine 5* for video generation.

**(3) Data Scaling Behavior.** Despite the strong scene diversity provided by numerous expert photography images, we observed that model outputs become unstable as the training data volume increases. As shown in Tab. 3c, Fig. 4a, and Fig. 11, while both accuracy and quality initially improve with increasing data volume, performance deteriorates when the training set size grows further. We hypothesize that this challenge lies in maintaining the desired model behavior to ensure proper perspective rather than deviating into unintended random behaviors. This phenomenon bears similarities to what was described in IC-Light [52], and we plan to incorporate their training methodology to investigate whether it can enhance training stability in our future work.

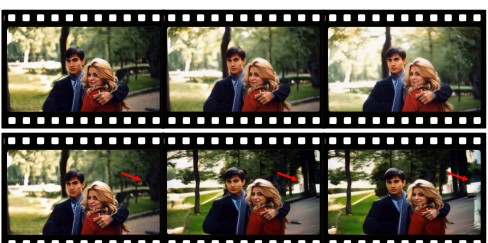

Figure 11: Diverse data presents instability.

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

# Appendix

## A  Prompt Instructions

For VQ and MQ, our instructions primarily follow those of VideoAlign [24]. We specifically designed the CA instructions as shown below:

> **Input Template (CA part)**
>
> **\*\*Composition Aesthetic\*\*:**
> Evaluate the evolution and sophistication of compositional techniques throughout the video, drawing inspiration from Magnum Photos' aesthetic principles. Consider the following sub-dimensions:
> - **\*\*Layering Complexity\*\***: Assess how the video utilizes multiple planes and creates depth through foreground, middle ground, and background interactions. Consider if these spatial relationships become more sophisticated over time.
> - **\*\*Geometric Harmony\*\***: Evaluate the use of strong geometric elements, lines, and shapes that create dynamic tension and visual interest, similar to Alex Webb's approach to complex frame organization.
> - Color Relationships: Consider how color blocks and contrast are used compositionally to create visual weight and guide viewer attention through the frame.
> - **\*\*Frame Utilization\*\***: Evaluate how effectively the entire frame is used, including edges and corners, and how secondary elements support the main subject.
> - **\*\*Visual Rhythm\*\***: Consider the pattern and repetition of elements, and how they create compositional flow and movement within the frame.
> - **\*\*Juxtaposition Development\*\***: Assess how the video develops and maintains meaningful visual relationships between different elements in the frame.
>
> Please provide the ratings of Composition Aesthetic: **<|CA_reward|>**
> END

**Instruction Source.** The compositional principles in our framework are derived from seminal works in photography theory and practice. These include the layering complexity theory from Alex Webb's "The Suffering of Light" and Sam Abell's three-layer composition approach; geometric harmony principles from Henri Cartier-Bresson's "The Decisive Moment"; color relationship theories from Steve McCurry and Ernst Haas; frame utilization techniques from Robert Frank's "The Americans"; and visual rhythm concepts from Paul Strand and Minor White. These principles, extensively documented in the works of Magnum Photos photographers, form the theoretical foundation for our composition assessment criteria.

Table 5: Key points summary outlined in annotation guidelines for CA evaluation dimension.

| Evaluation Dimension | Key Points Summary |
|---|---|
| Composition Aesthetic | Considering the following dimensions in the compositional design of the video:
- **Compositional Reasonableness**: The composition should be objectively reasonable and well-balanced.
- **Compositional Clarity**: The arrangement of elements should be clear and visually organized.
- **Compositional Detail**: The level of sophistication in the arrangement and relationship between elements.
- **Compositional Creativity**: The composition should be aesthetically pleasing and show creative arrangement.
- **Compositional Safety**: The composition should not create visual tension or uncomfortable viewing experience. |

