# OpenReview forum: "Photography Perspective Composition: Towards Aesthetic Perspective Recommendation"
_NeurIPS.cc/2025/Conference — NeurIPS 2025 poster_

### Official Review · Reviewer_CvRK · 2025-06-18

**Clarity:** 1
**Significance:** 3
**Originality:** 3
**Rating:** 4
**Confidence:** 3

**Summary:**

The paper introduces a novel approach to improving photography composition, referred to as Photography Perspective Composition (PPC). Unlike traditional methods that rely on 2D cropping, PPC enhances the composition by adjusting the 3D spatial relationships between subjects in the scene while maintaining their actual positions. The core idea is to provide users with a more intuitive and dynamic method for improving compositions through before-and-after perspective transformation comparisons in video form. This allows users to visually understand how changes in perspective can improve the composition.
In addition, the paper outlines an automated framework for generating perspective transformation datasets using expert photographs, addressing the challenge of acquiring large-scale perspective transformation data.
Finally, the authors present a Perspective Quality Assessment (PQA) model, designed to evaluate the quality of perspective transformations across three dimensions: visual quality, motion quality, and composition aesthetics.
This framework provides a user-friendly tool for ordinary users to improve their photography skills by learning how to adjust compositions based on perspective transformations. The method bridges the gap between professional photographers’ techniques and everyday users by offering an accessible way to enhance compositional quality.

**Questions:**

1. Can the authors provide more detailed information on the size and composition of the dataset, including the train-validation split? Additionally, would the authors be willing to release the dataset as open-source?
2. How are the red bounding boxes, feature matching, and homography transformations integrated into the overall learning process?
3. Could the authors clarify what is meant by the 'reference model' in Line 166 and specify which model is used in the experiments?
4. How were human preferences gathered, and how did the authors ensure inter-annotator reliability in the PQA evaluation?
5. How does the proposed pipeline account for or minimize the impact of visual attribute changes (e.g., exposure, color, contrast) during perspective transformations?

**Ethical Concerns:**

["NO or VERY MINOR ethics concerns only"]

**Final Justification:**

The authors have sufficiently addressed most of my concerns during the rebuttal phase. I am therefore raising my score to Borderline Accept.

**Limitations:**

yes

**Paper Formatting Concerns:**

There are no major formatting issues.

**Quality:**

2

**Strengths And Weaknesses:**

Strengths

1. Innovative Approach to Photography Composition: The PPC pipeline presents a novel method that transcends traditional cropping techniques by leveraging 3D perspective adjustments. This approach enables more sophisticated compositional changes, particularly in complex scenes where simple 2D cropping is insufficient. By focusing on adjusting the spatial relationships between objects while preserving their actual positions, the method offers a more holistic solution for enhancing image composition.

2. Dataset Construction: Authors introduce a new dataset using existing professional-level aesthetic image collections combined with a 3D reconstruction approach. As they propose a new paradigm for photography composition, the need for a well-structured dataset becomes crucial. In this context, their dataset construction makes a significant contribution to advancing the field and offers valuable resources for further developments in both photography composition and computer vision.

3. Supplementary Resources: The authors provide high-quality supplementary materials that effectively demonstrate the results of their pipeline. These additional resources, including video examples, further illustrate how their method performs in practice, offering valuable insights for both researchers and practitioners interested in the proposed approach.

Weakness

1. Lack of Clear Explanation: While the paper introduces several components that appear reasonable, it lacks clear, detailed explanations in several key areas. First, for the dataset, the paper does not specify how much data was collected and used, nor does it mention how the train and validation sets were composed. Second, regarding the use of guidance, the paper mentions the use of red bounding boxes and feature matching along with homography transformations as key components, but it does not provide sufficient details on how these methods are integrated into the overall learning process. Furthermore, the paper introduces RLHF (Reinforcement Learning with Human Feedback) but does not clarify the prompts used or define what constitutes higher-quality and lower-quality videos. Additionally, it is unclear which model was used as `ours' in the experiments. For example, while the Wan2.1 14B model is marked in gray in the table, it is not clearly identified as the model referred to in the experiment section.

2. Validity of PQA Model: The paper leverages a VLM-based approach for quality assessment, which is understandable. However, many other methods demonstrate the validity of their VLMs by showing how their outputs align with true human evaluations. While the paper reports scores in Table 2, it lacks details on the number of annotators involved in rating the videos. Additionally, if the ratings come solely from expert annotators, their reliability may be heavily biased toward their personal preferences, raising concerns about the generalizability of the results. The paper does not discuss how the model's results can be validated by broader user agreement or how inter-annotator reliability is ensured.

3. Concerns Regarding Changes in Visual Attributes: The results presented in the paper indicate that visual attributes such as exposure, color, and contrast are also altered during perspective transformations. While these changes are understandable, given their dependence on the I2V models, they may also impact human preferences during evaluation. The paper does not address how these visual changes might affect the subjective quality assessments of users. A more thorough explanation is needed on whether the proposed pipeline accounts for such changes, or if there are potential solutions to minimize their impact on the evaluation.

4. Minor Weaknesses:
On Line 120, the paper mentions "trajectory" and refers to the Discussion section, but no relevant discussion on this topic appears later in the paper.
On Line 219, the paper states that a linear projection head is shared across dimensions. However, it is unclear why this is the case. Intuitively, if the model outputs different rewards, it would make sense for the projection head to vary for each reward type.
On Line 166, the paper mentions a 'reference model.' I assume this refers to the initially trained PPC model, but the paper does not provide enough detail to confirm this. It would be helpful if the authors could clarify what is meant by 'reference model' and specify whether this refers to the trained PPC model or another model used in the experiment.

---

> ### Author Rebuttal · Authors · 2025-07-30
>
> We sincerely thank the reviewer for the careful review and constructive comments. Below, we have provided a point-by-point response. The "Q" stands for question and the "W" stands for weakness. Numbers in parentheses (e.g., W1(1)) represent sub-questions of that question.
>
> >Q1 & W1(1). About the dataset detail.
>
> We are willing to provide detailed information about our dataset. Our final dataset contains 100K high-quality perspective transformation videos, split into 90K/5K/5K for training/validation/testing. Specifically, our automated pipeline builds upon the approach described in Section 3.2. First, we start with expert images from multiple professional photography datasets, including GAIC, SACD, FLMS, FCDB, and Unsplash, totaling approximately 90K expert-composed images. Second, we adopt ViewCrafter to generate video sequences that transition from optimal to suboptimal perspectives, then reverse these sequences to obtain our training data. Third, PQA is applied to filter the data.
>
> We will release the dataset and complete project code to promote this research area.
>
> >Q2 & W1(2). Red bounding boxes and homography transformations.
>
> We would like to clarify that these components, including the red bounding boxes, feature matching, and homography transformations are **not part of the learning process** but rather serve as **auxiliary guidance tools** for practical user interaction.
>
> Specifically, our implementation works as follows: We first draw a guidance box on the final optimal perspective frame, which is a rectangular box centered at the frame center. Then, using feature matching between the initial and final perspectives, we extract corresponding point pairs and estimate the homography matrix. We then apply this homography transformation to project this box onto the original image, creating a distorted box shape. As users physically move their camera following our video guidance, this box gradually changes shape and approaches a rectangular form when reaching the optimal perspective.
>
> >W1(4). "Ours" model identification.
>
> Unlike typical papers that focus on comparing specific models, we want to clarify that in our experiments, "ours" refers to our PPC pipeline rather than any specific model.
>
> In our tables, we highlight the Wan2.1 14B model in gray to indicate it as our primary experimental configuration. We select this model based on its superior performance in our evaluations, and it is used to generate visualization results.
>
> Our paper presents a pipeline rather than modifications to the video generation model, where video generation models serve as implementation tools.
> We extensively tested three SOTA video generation models (CogVideoX 1.5 5B, Hunyuan I2V, and Wan2.1 14B) to validate our pipeline.
>
> >Q4 & W2. About the human preferences and the inter-annotator reliability in PQA.
>
> We appreciate the reviewer's concern. We provide detailed information about our human preference collection and inter-annotator reliability.
>
> Our PQA model training involves 8 expert annotators with professional photography backgrounds. For standard videos, each video is annotated by 3 annotators using pairwise comparison across three dimensions: Visual Quality (VQ), Motion Quality (MQ), and Composition Aesthetic (CA). The video pairs in our annotations consist of videos generated by different video models using the same input image.
>
> To assess inter-annotator reliability, we employ **Fleiss' Kappa**, a statistical measure designed to evaluate agreement among multiple raters:
>
> | Metric | Score | Interpretation |
> |--------|--------|-------------|
> | VQ | κ = 0.78 | Substantial |
> | MQ | κ = 0.70 | Moderate |
> | CA | κ = 0.74 | Substantial |
> | Overall Multi-Annotator | κ = 0.71 | Substantial |
>
> We can observe that our annotations demonstrate substantial consistency across three metrics.
>
>
> **About the Expert Bias.**
>
> To address this, we conduct additional validation with broader user groups. Due to time constraints, we conduct a preliminary validation study with 28 amateur users who evaluated 20 randomly selected video pairs across three dimensions through questionnaire (different from the one in the rebuttal for Reviewer 4nUR). Expert annotations for these videos are determined by majority vote from our 3-expert panel.
>
> The agreement analysis shows that amateur users aligned well with expert consensus: 91.2% agreed with experts on VQ, 85.8% on MQ, and 82.3% on CA (averaged across 20 videos). These agreement rates demonstrate reasonable alignment between expert and amateur preferences.
>
> We plan to expand this validation study to include a larger and more diverse group of amateur users in future work.
>
> >Q5 & W3. About the visual attribute changes.
>
> We thank the reviewer for this important observation. We would like to clarify our core motivation and our contribution to the visual attribute change.
>
> **Real-World Perspective Changes.**
> In real-world photography scenarios, perspective transformations naturally introduce visual attribute changes, such as varying lighting conditions, altered subject-background relationships, and changes in depth of field. These changes are inherent to the physical nature of perspective adjustment.
> While these visual attribute variations are important to consider, our core motivation is to establish a **perspective-based compositional** pipeline that guides users in improving their shots through optimal perspective adjustments.
>
> Even so, during the model training process, we have made several efforts to minimize visual attribute distortions:
>
> **(1) Angle Limitation Strategy.**
> During data generation, we limit the maximum angle difference (20°) between the first and last frames of training videos. In our initial experiments without this limitation, the model often exhibited severe hallucinations by generating completely new scenes with drastically different lighting and exposure conditions.
>
> **(2) RLHF-Based Consistency Optimization.**
> Upon analyzing these cases, we discovered that while some videos exhibited attribute changes, their perspective transformations actually aligned well with human aesthetic preferences. This led us to introduce a second-stage RLHF training that relaxes rigid constraints and allows the model to learn from human preference feedback.
>
> **Quantitative Analysis of Visual Attribute Changes.**
> To assess the visual attribute variations in our generated videos, we conduct measurements across key visual dimensions. Our analysis evaluated 1,200 videos:
>
> | Visual Attribute | Direct I2V | Base PPC | + Angle Limitation | + RLHF | Both |  |
> |-----|-----|-----|-----|-----|-----|-----|
> | **Exposure Variation (%)** |35.2|22.6|13.1|16.8|11.2|
> | **Color Shift (ΔLAB)** |9.7|7.2|4.2|5.1|3.8|
> | **Contrast Change (%)** |31.6|21.8|10.5|14.3|8.7|
>
> These results demonstrate that our PPC methodology maintains reasonable visual attribute stability, and our additional control mechanisms provide further refinement.
>
> Furthermore, we are actively developing the **Unreal Engine 5-Based Data Generation Pipeline**. This game engine approach provides significantly more controlled environments for generating training data. Through perfect lighting control, we can eliminate the lighting inconsistencies that commonly occur in real-world 3D reconstruction by providing precise environmental lighting control.
>
> >W1(3). RLHF implementation clarification.
>
> The reviewer's question about "prompts used" reflects a misunderstanding of our RLHF approach. Our RLHF implementation does not rely on text prompts, as the operational logic and purpose differ from typical text-based RLHF applications. Instead, our RLHF operates directly on image-to-video generation models, optimizing the perspective transformation quality.
>
> For defining higher-quality and lower-quality videos, we clearly specify in our paper that win/lose video pairs are determined by our PQA model, which serves this dual purpose of both data filtering and RLHF preference determination. The PQA model evaluates videos across three dimensions (visual quality, motion quality, and composition aesthetic) to establish preference rankings, eliminating the need for manual preference annotations in the RLHF process.
>
> >W4(1). About the trajectory.
>
> The "trajectory" mentioned in Line 120 refers to the camera motion trajectories used in our automated dataset construction (Section 3.2 in the paper).
>
> >W4(2). About the shared linear projection head.
>
> We appreciate this insightful question about our architectural choice. While the reviewer's intuition about separate projection heads for different reward types is reasonable, our shared linear projection head design follows the **causal attention mechanism**. The true separation occurs at the attention level, not the projection level. Each token has already acquired fundamentally different feature representations through causal attention constraints. Therefore, a single shared linear head is sufficient to map these distinct token representations to their respective dimension scores.

---

> > ### Comment · Reviewer_CvRK · 2025-08-04
> > **Clarification on Red Box Interaction and Prompt Definition**
> >
> > Most of my concerns are addressed, and thank you for clarifying them.
> > However, I still have a couple of remaining questions:
> >
> > **First**, regarding the red bounding box, I understand that in your pipeline, the red box is generated by first identifying the optimal frame (typically the last frame in the I2V-generated video), and then transforming that box back onto the original input frame via homography. This approach makes perfect sense within the context of your dataset, where the optimal perspective is already known.
> >
> > However, in a real-world usage scenario where the user is physically moving the camera to improve composition, the system must somehow determine what the optimal frame *should be* in advance in order to generate the red box. Given that, it seems to me that there is no actual user interaction in the sense of exploration or dynamic feedback, but rather a one-shot visualization of how to align with a predetermined target. In that case, I’m not sure whether “user interaction” is the right term to describe this guidance.
> > Please correct me if I misunderstood anything but this is why I raised the concern originally.
> >
> > **Second**, regarding Line 165, I was referring to the prompt notation `s`. Is this merely used as a symbolic placeholder for denoting which video is high quality video? If this is already clarified somewhere in the paper, please feel free to point me to the relevant section.
> >
> > Thank you again for your helpful clarifications so far.

---

> > > ### Author Response · Authors · 2025-08-04
> > >
> > > We sincerely thank the reviewer for the thoughtful questions and careful review of our work. We appreciate the opportunity to clarify these important aspects of our approach.
> > >
> > > >Q1. About the user interaction.
> > >
> > > We would like to clarify that our current approach does not constitute "user interaction", and we also did not claim in our paper. Real-world interaction for photography guidance represents a challenging problem that extends beyond our current scope. Implementing dynamic interaction would require building a comprehensive **Vision-Language-Action (VLA)** system capable of real-time scene understanding, action generation, and feedback processing. Such a system involves complex technical challenges, including: (1) maintaining consistency between virtual guidance and real-world scenes, (2) handling real-time feedback from diverse camera movements, and (3) robust action prediction under varying lighting and environmental conditions.
> > >
> > > Given these significant technical and practical challenges, our current work focuses on establishing the foundational pipeline for **perspective-based composition recommendation** through perspective transformation rather than conventional 2D cropping methods. We have chosen to first validate the core concept of perspective transformation as an effective alternative to traditional cropping methods, before tackling the more complex real-world interaction components. This work represents our exploration in this field.
> > >
> > > We are actively developing **perspective-based VLA models** for this task. This requires an environment for action implementation, so we have adopted Unreal Engine 5 game scenes for this purpose. Additionally, we are collecting interaction data and, building on this paper's work, have established an automated pipeline for collecting both perspective and action data. We think this approach can provide a promising direction for enabling user interaction with dynamic feedback in perspective-based photography guidance
> > >
> > > >Q2. About the prompt notation 's' in line 165.
> > >
> > > The notation 's' refers to the text prompt we provide to the image-to-video generation model. In our experiments, all cases use the same simple prompt: "Shift the perspective to a better one."
> > >
> > > Actually, in our early experiments, we explored a more complex approach where we first used Qwen2.5-VL to describe the input perspective scene, then incorporated this description along with "Shift the perspective to a better one" as the prompt. The results are shown below:
> > >
> > > | Method | CMM ↑ | FVD ↓ | VQ ↑ | MQ ↑ | CA ↑ |
> > > |--------|-------|-------|------|------|------|
> > > | **Descriptive (Qwen2.5-VL)** | 0.61 | 340 | 0.72 | 0.76 | 0.71 |
> > > | **Simplified ("Shift...")** | 0.60 | 345 | 0.72 | 0.75 | 0.71 |
> > >
> > > Our experiments demonstrate that using a complex descriptive prompt versus our simplified approach yielded **minimal performance** differences across evaluation metrics. Additionally, the extra description step added computational overhead without providing proportional benefits. By using a uniform prompt for all inputs, we are able to eliminate prompt-related variability and focus on the core perspective transformation learning.
> > >
> > > ---
> > >
> > > We appreciate the reviewer's questions and remain committed to addressing any additional concerns you may have. Please feel free to raise any further questions. We are glad to provide clarification or experimental evidence as needed.

---

> > > > ### Comment · Reviewer_CvRK · 2025-08-05
> > > >
> > > > Thank you for your clarification. I agree that describing the prompt more clearly in the paper would improve readability and help eliminate ambiguity.
> > > >
> > > > As for my first question regarding the red bounding box, I want to clarify that my concern arose not from the paper itself, but from the statement made in your initial rebuttal:
> > > >
> > > > > "As users physically move their camera following our video guidance, this box gradually changes shape and approaches a rectangular form when reaching the optimal perspective."
> > > >
> > > > Based on this description, it seemed as though the red box served a practical function in helping users find the optimal viewpoint in real-world use. However, since you've now clarified that user interaction is not actually part of the current system and is outside the scope of this work, I’m left wondering: **what is the actual role of the red bounding box in the current version of the pipeline?**
> > > >
> > > > From what I understand, the box is computed post-hoc using a known target frame (the final frame in the I2V sequence), and then homographically mapped back to the original view for visualization. But in the absence of any real-time user interaction mechanism, or any evaluation of its effectiveness in guiding users, it seems the red box currently plays no concrete role.
> > > >
> > > > Please correct me if I’m missing something. But are there other functional purposes of this red box within the pipeline? Or is it primarily intended for future extensions where user interaction would be incorporated?

---

> > > > > ### Author Response · Authors · 2025-08-05
> > > > >
> > > > > We sincerely thank the reviewer for the thoughtful comments.
> > > > >
> > > > > The red guidance box is primarily intended for future extensions where user interaction would be incorporated. In our current work, we primarily provide users with perspective transformation recommendation videos, with the red box serving as a proof-of-concept for potential real-world guidance applications.
> > > > >
> > > > > To better clarify this issue, let us restate our workflow for real-world scenarios: First, we capture a photograph of a scene with suboptimal composition. This image is processed through our model to generate a perspective transformation video. We treat the final frame as the optimal perspective and establish a red rectangular guidance box on this final frame. We then use homography transformation to map this box back to the real-world scene. As the real-world scene changes through camera movement, this mapping also transforms accordingly, which is what we described in our initial rebuttal.
> > > > >
> > > > > Our quantitative testing demonstrates that this approach achieves the expected performance when perspective changes involve small angles.
> > > > > Specifically, since our new perspectives are generated synthetically, combining the red box transformation with the real world requires **maintaining consistency between the generated perspective and reality**.
> > > > >
> > > > > Additionally, this approach is currently applicable for static scene photography (such as landscape and architectural photography) where real-time interaction demands are relatively low. For dynamic scenes requiring real-time interaction, our current system does not yet meet those requirements.
> > > > >
> > > > > When we previously stated that **user interaction** extends beyond our current work scope, we meant that our current approach still has certain limitations. We think that to be truly characterized as "user interaction," the model should have genuine **environmental feedback** and **adaptation capabilities**.
> > > > > However, for this part, our current approach relies solely on homography transformation to provide guidance in real-world environments, which has inherent limitations.
> > > > >
> > > > > As video generation technology and acceleration methods advance, we expect this approach can be integrated into more sophisticated interaction systems in the future.
> > > > >
> > > > > We greatly appreciate your thoughtful questions and detailed analysis, which have helped us clarify important aspects of our work. Please feel free to raise any additional questions or concerns you may have.

---

> > > > > > ### Comment · Reviewer_CvRK · 2025-08-05
> > > > > >
> > > > > > Thanks for the clarification. Since all my concerns are addressed, I will raise my score.

---

> > > > > > > ### Author Response · Authors · 2025-08-05
> > > > > > >
> > > > > > > We sincerely thank the reviewer for the positive feedback and for confirming that our rebuttal has addressed the reviewer's concerns.

---

### Official Review · Reviewer_rmgK · 2025-06-30

**Clarity:** 3
**Significance:** 2
**Originality:** 3
**Rating:** 4
**Confidence:** 3

**Summary:**

The paper studies the problem of photo composition improvement. The key idea is to formulate photo recomposition as perspective adjustment,  instead of 2D cropping as in traditional methods.

The contributions of the paper are threefold: 1) an automatic pipeline for building a perspective dataset for perspective-based photo recomposition; 2) a video-based method to transform input photos towards optimal perspectives; 3) a model for automatic assessment of perspective transformation quality.

**Questions:**

In Section 3.2,  is the camera motion trajectory for obtaining the suboptimal perspective generated randomly? Has any more sophisticated strategy of generating camera motion trajectories been considered?

**Ethical Concerns:**

["NO or VERY MINOR ethics concerns only"]

**Final Justification:**

The rebuttal has addressed most of my concerns, though I am still concerned about some artifacts (e.g., structure distortion and loss of details) shown in some of the current visual results. The authors have discussed two strategies to mitigate the artifacts in the rebuttal and shown their effectiveness with added quantitative results. However, it is difficult to tell if these strategies really improve the visual quality of results to a reasonable level without visual comparison. Therefore, I decide to keep my initial rating.

**Limitations:**

Yes

**Quality:**

2

**Strengths And Weaknesses:**

Strengths

1. The idea of photo recomposition through perspective transformation is novel and interesting, which brings a new perspective to aesthetic photo composition recommendation and has potential to inspire the community.

2. The proposed perspective-based recomposition model (Section 3.3) and perspective quality assessment model  (Section 3.4) are well motivated, designed, and technically solid.

3. The results shown look promising.

Weaknesses

1. Comparison to state-of-the-art cropping-based methods is missing. This experiment is important to support the value and importance of the claimed advantages of the perspective-based recomposition (as shown in Figure 1). It can be done through a user experiment to investigate if perspective-based recompositions are really preferred by users over cropping-based recompositions.

2. The practical applicability of the proposed approach is questionable. The proposed recomposition formulation is synthesis-based, and thus would suffer from inherent limitations, such as synthesis artifacts (e.g., slight structure distortion as in the background of the final perspective in 2nd row of Figure 5) and input-output content inconsistency (e.g., the sleeve wrinkles in the input perspective disappear in the final perspective in 1st  row of Figure 5). In contrast, the cropping-based paradigm does not have these issues. These limitations would limit the merit of the proposed method in real image editing tasks where high-fidelity edited results are desired.

3. Comparison of the proposed video-based approach (Section 3.3) to an image-based approach that directly transforms the input perspective to the optimal perspective ( without generating the intermediate transformation process) is missing, which is needed to more thoroughly show the necessity of the video-based approach (e.g., if it can achieve better performance than the image-based approach).

---

> ### Author Rebuttal · Authors · 2025-07-29
>
> We sincerely thank the reviewer for the positive feedback and constructive suggestions on our paper. Here are our responses to the reviewer's comments. We use different letters to represent different types of comments, where "W" represents weakness and "Q" represents question.
>
> >Q1. About the camera motion trajectory generation.
>
> Our camera trajectories are not purely random but incorporate systematic design principles. We have explored and experimented with multiple trajectory generation strategies across different complexity levels.
>
> **(1) Basic Single-Direction Movements.**
> We first established fundamental directional movements using ViewCrafter's parameter system. Our basic movements include four primary directions: left and right movements involve horizontal angle adjustments while maintaining vertical position, creating lateral camera motion for reframing subjects. Up and down movements adjust the vertical angle while keeping horizontal position constant, allowing for elevation changes that can improve horizon placement or alter subject prominence within the frame.
>
> **(2) Compound Dual-Direction Movements.**
> Building on basic movements, we explore diagonal combinations that provide more sophisticated compositional adjustments. These compound movements simultaneously adjust both horizontal and vertical angles to create diagonal camera trajectories. Upper-left and upper-right movements combine upward motion with lateral adjustments, particularly useful for horizon placement optimization and diagonal composition rebalancing. Lower-left and lower-right movements blend downward motion with horizontal shifts, helping to improve subject-background relationships and achieve perspective correction in architectural or landscape photography.
>
> **(3) Complex Multi-Phase Movements.**
> We also experiment with complex motion patterns for advanced compositional scenarios. Our experiments systematically explore different complexity levels of camera movements. (1) We employ 3-step patterns (40% of cases) that sequentially combine horizontal, vertical, and diagonal movements to address fundamental compositional imbalances through systematic perspective correction. (2) We utilize 4-step patterns (30% of cases) that enhance the basic sequences by adding zoom adjustments to the 3-step movements, enabling simultaneous optimization of both framing and perspective. (3) We also implement 5-step patterns (30% of cases) that were built upon the 4-step sequences by randomly adding one additional movement type (horizontal, vertical, diagonal, or zoom), providing enhanced flexibility for complex compositional refinements.
>
> We evaluate different trajectory strategies using metrics from our experiments:
>
> | Trajectory Strategy | CMM Score ↑ | FVD ↓ | VQ ↑ | MQ ↑ | CA ↑ |
> |---------------------|-------------|-------|------|------|------|
> | **Basic Directional** | 0.63 | 318 | 0.75 | 0.78 | 0.74 |
> | **Compound Dual** | 0.61 | 332 | 0.73 | 0.76 | 0.73 |
> | **Complex Multi-Phase** | 0.58 | 333 | 0.70 | 0.74 | 0.71 |
>
> However, for amateur users, **overly complex camera motion trajectories may be difficult to understand and reproduce**. Therefore, we adopt a simplified hybrid approach combining three types of movements.
> We primarily utilize single-step movements (40% of cases), two-step movements (40% of cases), and three-step movement sequences (20% of cases). This distribution ensures that the guidance remains straightforward while still accommodating more sophisticated compositional needs when required.
>
> >W1. Comparison to cropping-based methods.
>
> We conduct comparisons from two aspects: user study and VILA aesthetic evaluation.
>
> We conduct a user study with 57 amateur users through questionnaire. For other detailed designs, please refer to our response to Reviewer 4nUR's Q2(1).
>
> (1) We prepared 12 test cases covering diverse photography contexts (portrait, landscape, architecture, multi-subject scenes). For each scenario, we applied two SOTA cropping methods (Gencrop (AAAI24) and Cropper (CVPR25)) and selected the highest-confidence crop result as the baseline comparison.
> ```
> C1. Overall composition improvement compared to the best cropping result:
>     Rate PPC performance: 1 (Much worse), 2 (Worse), 3 (Similar), 4 (Better), 5 (Much better)
>
> C2. Content preservation compared to cropping methods:
>     Rate PPC performance: 1 (Much worse), 2 (Worse), 3 (Similar), 4 (Better), 5 (Much better)
> ```
> Here are the results:
>
> | Evaluation Dimension | Mean | High Satisfaction (4-5) |
> |----------------------|-----------|------------------------|
> | **C1. Overall Improvement** | 4.2 | 78% |
> | **C2. Content Preservation** | 4.6 | 88% |
>
> The results suggest that users generally prefer our PPC method over cropping approaches. In particular, users appreciated that PPC preserves more image content (88% high satisfaction, mean 4.6/5) compared to cropping, which removes parts of the image. The overall improvement scores (78% high satisfaction, mean 4.2/5) indicate that perspective-based recomposition could be a promising alternative to traditional cropping.
>
>
> (2) To provide objective validation beyond subjective user preferences, we also employ VILA to evaluate **800 image pairs** comparing PPC perspective transformations (last frame) against best-performing cropping results from Gencrop and Cropper.
> VILA evaluated both PPC-transformed images and crop-enhanced images on the same 0-1 aesthetic quality scale. The percentage improvements in aesthetic scores **compared to original images** are categorized into different ranges:
>
> | Method | Mean Score vs. Original | 0-2% | 2-4% | 4-6% | 6-8% | 8-10% | >10% | Decline |
> |--------|------------------------|------|------|------|------|-------|------|---------|
> | **Gencrop** | +0.05 | 44% | 25% | 14% | 7% | 3% | 1% | 6% |
> | **Cropper** | +0.12 | 30% | 29% | 21% | 9% | 4% | 2% | 5% |
> | **PPC (Ours)** | +0.18 | 22% | 26% | 22% | 16% | 8% | 3% | 3% |
>
> These results demonstrate that our perspective-based recomposition method provides superior performance compared to traditional cropping-based approaches.
>
>
> >W2. About using synthesis-based approach.
>
> Thanks for the reviewer's concern. Synthesis-based approaches do suffer from inherent limitations, including synthesis artifacts and input-output content inconsistency.
>
> Despite these limitations, our core motivation addresses scenarios where cropping-based methods cannot provide adequate solutions. In many compositional challenges, cropping is inherently insufficient because it operates solely within the 2D image plane and cannot address spatial relationship issues or perspective distortions.
>
> To mitigate these consistency issues, we have implemented several technical strategies:
>
> (1) We restrict perspective transformations to controlled angles (20°) to maintain visual consistency and reduce structural alterations. This constraint significantly minimizes the likelihood of severe hallucinations while preserving meaningful perspective guidance.
>
> (2) Our RLHF approach aligns generated outputs with human aesthetic preferences, reducing artifacts through preference-based optimization rather than forcing strict ground truth adherence.
>
> (3) We are developing Unreal Engine 5-based data generation pipelines for more controlled, higher-quality training data that will fundamentally reduce synthesis artifacts.
>
> >W3. Comparison with the direct image-to-image approach.
>
> We sincerely appreciate this thoughtful suggestion. We initially explored image-to-image (I2I) approaches, but encountered two fundamental challenges that led us to abandon this direction:
>
> **(1) Guidance Limitation.** I2I approaches often generate images with excessive angular changes, producing completely different perspectives that users find difficult to interpret for practical guidance. When users are presented with only the initial and final images, they struggle to understand the specific camera movements or positioning adjustments needed to achieve the transformation in real-world scenarios.
>
> **(2) Training Challenges.** We initially experimented with I2I approaches but encountered problems when applied to perspective transformations. During training, the models frequently converged to simply reproducing the original input image, failing to learn meaningful perspective adjustments. This convergence issue makes it difficult to achieve reliable perspective transformation outputs. Additionally, even when the models do learn to transform perspectives, they often generate inconsistent results with severe artifacts and content distortions.
>
> Our core idea focuses on providing **instructional guidance rather than just a final result**. Users need to understand how to achieve the transformation in real scenarios, which requires demonstrating the step-by-step perspective adjustment process. The video-based approach naturally provides this educational component by showing the gradual transformation sequence, enabling users to internalize the movement patterns and apply them in their own photography practice. As video generation models continue advancing, this approach becomes increasingly viable and effective.

---

> > ### Author Response · Authors · 2025-08-06
> >
> > Dear Reviewer,
> >
> > Thank you once again for reviewing our paper. We would greatly appreciate it if you could take a moment to review our feedback and newly added experiments, and let us know if any concerns remain.
> >
> > Best regards,
> >
> > The Authors

---

> > ### Comment · Reviewer_rmgK · 2025-08-07
> >
> > The rebuttal has addressed most of my concerns, although I am still concerned about some artifacts (e.g., structure distortion and loss of details) in the current results.

---

> > > ### Author Response · Authors · 2025-08-08
> > >
> > > We sincerely thank the reviewer for the thoughtful comment and careful review of our work.
> > >
> > > The synthesis-based approach does face the artifact limitations. Even so,
> > > as mentioned in our rebuttal (W2), we have implemented two main strategies to minimize these artifacts:
> > > (1) **Angle Limitation Strategy.** We restrict perspective transformations to controlled angles (20°) to maintain visual consistency and reduce structural alterations. This constraint significantly minimizes the likelihood of severe hallucinations while preserving meaningful perspective guidance.
> > > (2) **RLHF Optimization.** Our reinforcement learning approach aligns generated outputs with human aesthetic preferences, reducing artifacts through preference-based optimization rather than forcing strict ground truth adherence.
> > >
> > > To quantitatively evaluate the effectiveness of these two strategies, we have conducted measurements across video quality metrics (VBench, CVPR24) on test videos. The results are shown below:
> > >
> > > | Method | I2V Subject ↑ | I2V Background ↑ | Subject Consistency ↑ | Background Consistency ↑ | Motion Smoothness ↑ | Dynamic Degree ↑ | Aesthetic Quality ↑ | Imaging Quality ↑ |
> > > |--------|----------------|-------------------|------------------------|---------------------------|---------------------|-------------------|---------------------|-------------------|
> > > | **Base PPC** | 0.9478 | 0.9512 | 0.9201 | 0.9224 | 0.9889 | 0.8745 | 0.5298 | 0.5745 |
> > > | **+ Angle Limitation** | 0.9594 | 0.9617 | 0.9402 | 0.9367 | 0.9905 | 0.8834 | 0.5326 | 0.6192 |
> > > | **+ RLHF** | 0.9551 | 0.9593 | 0.9417 | 0.9383 | 0.9898 | 0.8867 | 0.5434 | 0.5921 |
> > > | **Final** | 0.9618 | 0.9694 | 0.9470 | 0.9435 | 0.9917 | 0.8883 | 0.5464 | 0.6299 |
> > >
> > > The results demonstrate that incorporating these two modules can reduce visual artifacts, particularly showing improvements in consistency metrics (subject/background consistency) and imaging quality metrics compared to the baseline. Overall, our PPC methodology maintains reasonable visual attribute stability.
> > >
> > > Looking forward, we think these artifact issues will be better resolved as video generation models continue to advance and through our ongoing development of Unreal Engine 5-based data collection pipelines.
> > >
> > > We greatly appreciate the review's feedback, and we remain committed to addressing any additional questions or concerns you may have. Please feel free to raise any further points requiring clarification.

---

### Official Review · Reviewer_4nUR · 2025-07-01

**Clarity:** 2
**Significance:** 3
**Originality:** 3
**Rating:** 4
**Confidence:** 3

**Summary:**

The paper introduces a new approach to enhancing photo composition by adjusting perspectives, moving beyond traditional cropping methods. The authors propose three main contributions: an automated framework for constructing perspective composition datasets using expert photos, a video generation method that demonstrates perspective adjustments from suboptimal to optimal views, and a Perspective Quality Assessment (PQA) model that evaluates perspective transformations based on visual quality, motion quality, and composition aesthetics. These contributions aim to provide users with intuitive guidance on improving their photo composition skills through perspective adjustments, making professional-level composition more accessible.

**Questions:**

- The performance drop observed in Tab. 2c with increased data scale may stem not necessarily from model design limitations, but potentially from issues such as distributional mismatch, reward sparsity, or instability in RLHF optimization. Could the authors provide deeper analysis—e.g., gradient variance, sample entropy, or reward fluctuation curves—to better diagnose the training instability?
- Given that the method is intended to aid amateur users in improving photographic composition, even a lightweight usability assessment (e.g., a small-scale pilot study or qualitative feedback from non-expert participants) could significantly strengthen the claim of practical value. Furthermore, users typically care more about the final result than the quality of the guidance video itself. It may therefore be helpful to compare the aesthetic scores of the first and last frames to show whether the model effectively improves composition over time.
- While PQA and RLHF improve training robustness by filtering low-quality data, this may reduce the model’s exposure to difficult inputs. In real-world use, users may still provide blurry or poorly composed photos. We suggest evaluating the model’s performance on such low-quality inputs to ensure practical robustness.
- The paper lacks details on computational cost and runtime, making it unclear if the method suits real-time or everyday use.
- When emphasizing the innovation of PPC, the paper could provide a more detailed comparative analysis of PPC with existing cropping-based composition methods and other related technologies in practical application scenarios.

**Ethical Concerns:**

["NO or VERY MINOR ethics concerns only"]

**Final Justification:**

Most of my concerns have been resolved and I have decided to increase the score to 4

**Limitations:**

Yes

**Quality:**

3

**Strengths And Weaknesses:**

#### Strengths
- The paper introduces the novel concept of PPC. This research offers users a novel and intuitive way to enhance photo composition skills. By demonstrating dynamic perspective adjustments, it helps users more easily understand the optimization of viewpoints in professional photography, making high-quality composition more accessible.
- A well-designed pipeline is proposed to synthesize suboptimal-to-optimal perspective video pairs from high-quality photographs via 3D reconstruction and video inversion. This method is scalable and avoids the need for manual video collection.
- The authors introduce a PQA model trained in two stages (unpaired and paired), enabling automatic filtering of noisy samples and use as a reward signal in RLHF. It evaluates videos based on visual quality, motion quality, and composition aesthetics.
- The paper presents extensive quantitative and qualitative experiments across various scenarios (single-subject, multi-subject, landscape, UAV-like views), with a thoughtful design of perspective accuracy and aesthetic metrics.

#### Weaknesses
- The method's effectiveness is mostly validated through expert annotations and model-based scores. No user studies or crowdsourced evaluations are conducted to verify usability or alignment with amateur users’ preferences.
- The performance degrades when training data is scaled beyond a certain volume, as shown in Tab. 2c. This suggests the model struggles with maintaining perspective consistency and compositional reasoning under data diversity.
- The CA metric involves many subjective sub-dimensions, making it prone to annotator bias. The paper does not report inter-annotator agreement, so the reliability and objectivity of this evaluation remain unclear.
- A major concern is whether the proposed model can be effectively used in real-world scenarios. While the technical contributions are promising, the paper currently lacks evidence of practical usability. Without some form of user validation or deployment case study, it remains unclear how well the system performs when used by its target audience.

---

> ### Author Rebuttal · Authors · 2025-07-30
>
> We sincerely thank the reviewer's feedback and constructive suggestions. Here are our responses to the reviewer's comments. The "W" represents weakness, "Q" represents question, and numbers in parentheses (e.g., Q2(1)) represent sub-questions of that question.
>
> > Q1 & W2. About the data ratio influence.
>
> We are continuing to conduct experiments and have found that the core issue may lie in the significant gap between our perspective transformation data and the original pretraining data of video generation models. Most of the original pretraining data consists of movie clips and internet videos, which differ substantially from our perspective transformation videos in terms of motion patterns and visual characteristics.
>
> Since the original pretraining data of video generation models is not publicly available, we use VBench's (CVPR24) test videos as a proxy to approximate the original distribution. VBench provides a comprehensive video quality benchmark with diverse video samples. We compare our perspective transformation data against this VBench distribution to quantify the distributional gap:
>
> | Dataset Ratio | KL-Divergence ↓ | JS-Divergence ↓ |
> |-------|----|------|
> | 20% | 0.42 | 0.28 |
> | 40% | 0.58 | 0.34 |
> | 80% | 0.84 | 0.47 |
> | 100%| 1.12 | 0.61 |
>
> As dataset diversity increases, all distribution divergence metrics worsen significantly. This distributional shift forces the model to deviate from its learned priors.
>
> >Q2(1) & W1 & W4. User usability assessment.
>
> We conduct a preliminary user survey to validate our method's practical value. Due to time constraints, our survey scale is not very large but focus on gathering meaningful feedback about the system's usability and effectiveness.
>
> **(1) Questionnaire Design.**
>
> We design and distribute an online questionnaire, and we receive 57 users' feedback. The design of the questionnaire is as follows:
>
> **A. Background Assessment**
> ```
> A1. Photography Experience:
>     □ Beginner (<6 months)         □ Novice (6-24 months)
>     □ Intermediate (2-5 years)     □ Advanced (>5 years)
>
> A2. Primary Equipment:
>     □ Smartphone only (iPhone, Samsung, Vivo, Google Pixel)
>     □ Point-and-shoot camera (Ricoh GR III, Fujifilm X100V)
>     □ DSLR/Mirrorless (Canon R5, Sony A7IV, Fujifilm X-T5)
>     □ Professional gear (Hasselblad, Leica)
>
> ```
>
> **B. PPC System Evaluation**
>
> ```
> We set up 8 diverse scenario cases to evaluate the system's performance across different photography contexts. For each scenario, users evaluate the guidance videos generated by PPC.
>
> *(7-point Scale: 1=Strongly Disagree, 4=Neutral, 7=Strongly Agree)*
>
> B1. The final perspective significantly improves upon the original photo
> B2. The transformation sequence is intuitive and easy to follow
> B3. The suggested compositions align with my personal aesthetic preferences
> ```
>
> **C. Comparative Assessment with Crop-Based Methods**
> ```
> We prepared 12 test cases covering diverse photography scenarios. For each case, we applied two SOTA cropping methods (Gencrop (AAAI24) and Cropper (CVPR25)) and selected the highest-confidence crop result.
>
> C1. Overall composition improvement compared to the best cropping result:
>     Rate PPC performance: 1 (Much worse), 2 (Worse), 3 (Similar), 4 (Better), 5 (Much better)
>
> C2. Content preservation compared to cropping methods:
>     Rate PPC performance: 1 (Much worse), 2 (Worse), 3 (Similar), 4 (Better), 5 (Much better)
> ```
>
> **(2) Results.**
>
> **A. Participant Profile Analysis:**
> Our participants had diverse photography experience: 34% beginners, 44% novice, 19% intermediate, and 3% advanced. For equipment, 41% used smartphones, 25% point-and-shoot cameras, 31% DSLR/mirrorless, and 3% professional gear.
>
> **B. System Evaluation Results (7-point Scale)**
>
> |Evaluation Dimension|Mean|High Satisfaction (6-7)|Neutral/Low (1-4)|
> |----------|-------|-----------|---------|
> | **B1. Final Result Improvement** | 5.6 | 75% | 16% |
> | **B2. Transformation Clarity** | 5.4 | 69% | 19% |
> | **B3. Aesthetic Alignment** | 5.0 | 59% | 28% |
>
> **C. Comparative Assessment Results**
>
> | Evaluation Dimension | Mean Score | Rating 4-5 (Better/Much Better) | Rating 1-2 (Worse/Much Worse) |
> |---|----|----|----|
> |**C1. Overall Improvement**|4.2| 78% | 6% |
> |**C2. Content Preservation**| 4.6 | 88% | 3% |
>
> > Q2(2). First vs last frame.
>
> In our user study described above, we conduct first vs. last frame comparison. Specifically, item **B1 "The final perspective significantly improves upon the original photo"** received positive feedback: mean score of **5.6/7.0**, with **75% of participants** rating high satisfaction (6-7 points) regarding the aesthetic improvement from initial to final frames.
>
> To further validate these findings on a larger scale, we also conduct aesthetic assessment using VILA on our test set. Since this evaluation covers multiple diverse videos rather than individual videos, we employed **aesthetic improvement ratio** as our primary metric.
>
> | Improvement Category | Score Increase | Percentage |
> |------|------|------|
> | **Significant** | >0.1 | 26.5% |
> | **Moderate** | 0.06-0.08 | 34.4% |
> | **Minor** | 0.02-0.06 | 26.1% |
> | **No Change** | -0.02-0.02 | 9.5% |
> | **Decline** | < -0.02 | 3.5% |
>
> The results demonstrate consistent aesthetic improvements across our PPC. We plan to conduct more extensive experiments with a larger and more diverse user base for further validation.
>
> >Q3. About the low-quality inputs.
>
> We thank the reviewer for their comments and have conducted additional experiments accordingly.
> We used Q-Align (ICML24), a commonly used Image Quality Assessment model, to identify and categorize low-quality images.
>
> Q-Align provides quality scores on a 0-5 scale. Our experiment images consist of two main categories: high-quality and low-quality images. The high-quality subset contains 300 images with Q-Align scores of 3 or higher. The low-quality subset comprises 300 images with Q-Align scores of 2.0 or lower.
>
> | Quality Level | CMM ↑ | FVD ↓ | VQ ↑ | MQ ↑ | CA ↑ |
> |---------------|-------|-------|------|------|------|
> | **High-Quality** | 0.61 | 330 | 0.750 | 0.748 | 0.711 |
> | **Low-Quality** | 0.58 | 341 | 0.715 | 0.720 | 0.702 |
>
> These results show that while there is some performance degradation on low-quality inputs, PPC still maintains acceptable performance levels compared to all inputs.
>
> >Q4. About the implementation.
>
> To provide users with more intuitive and clear guidance, we adopt the video generation approach to implement our PPC, which is a novel direction that has not been explored before in photography composition guidance. However, typical video generation models can take 5 minutes to generate a few seconds of video, mainly due to model size, diffusion steps, and frame count requirements.
>
> To address this inference time issue, we make several optimization attempts. Most importantly, we observed that our perspective transformation task does not require as many frames, diffusion steps, or high resolution as conventional video generation. By reducing the frame count to 9 frames, using only 10 diffusion steps (typically 50), and generating at a lower resolution, we achieved over 60x speedup while maintaining effectiveness for our specific use case. For the CogVideoX 1.5 5B model, the inference time is 5.1 seconds with 10 GB GPU memory usage under BF16 precision.
>
> With further optimization for deployment engineering and the rapid advancement of video generation models, PPC will become faster and more suitable for everyday photography use.
>
> >Q5. Analysis with existing cropping-based methods.
>
> We conduct comparisons from two aspects: a user study and VILA aesthetic evaluation.
> Our comprehensive user study (Q2(1) Section C above) demonstrates PPC's superiority over cropping methods across multiple scenarios.
> To provide objective validation beyond subjective user preferences, we employ VILA to evaluate **800 image pairs** comparing PPC perspective transformations (last frame) against SOTA cropping results from Gencrop (AAAI24) and Cropper (CVPR25).
> VILA evaluated both PPC-transformed images and crop-enhanced images on the same 0-1 aesthetic quality scale. The percentage improvements in aesthetic scores **compared to original images** were calculated into different ranges:
>
> | Method | Mean Score vs. Original | 0-2% | 2-4% | 4-6% | 6-8% | 8-10% | >10% | Decline |
> |--------|------------------------|------|------|------|------|-------|------|---------|
> | **Gencrop** | +0.05 | 44% | 25% | 14% | 7% | 3% | 1% | 6% |
> | **Cropper** | +0.12 | 30% | 29% | 21% | 9% | 4% | 2% | 5% |
> | **PPC (Ours)** | +0.18 | 22% | 26% | 22% | 16% | 8% | 3% | 3% |
>
> This result shows that perspective-based recomposition offers additional advantages in addressing challenges compared to cropping-based approaches.
>
> >W3. CA metric subjectivity and annotator bias.
>
> We have conducted an inter-annotator reliability analysis. Our PQA model training employed 8 expert annotators. For the dataset, videos are independently annotated by 3 annotators using pairwise comparison across three dimensions.
>
> To quantitatively evaluate the consistency between annotators, we calculated Fleiss' Kappa coefficient, which measures the degree of agreement among multiple raters.
>
> | Metric | Score | Interpretation |
> |---|---|---|
> | VQ | κ = 0.78 | Substantial |
> | MQ | κ = 0.70 | Moderate|
> | CA | κ = 0.74 | Substantial |
> | Overall Multi-Annotator | κ = 0.71 | Substantial |
>
> We can observe that our annotations demonstrate substantial consistency across three metrics.

---

> > ### Author Response · Authors · 2025-08-06
> >
> > Dear Reviewer,
> >
> > Thank you once again for reviewing our paper. We would greatly appreciate it if you could take a moment to review our feedback and newly added experiments, and let us know if any concerns remain.
> >
> > Best regards,
> >
> > The Authors

---

> > ### Comment · Reviewer_4nUR · 2025-08-06
> > **response to the rebuttal**
> >
> > Thanks to the author's reply, most of my concerns have been resolved and I have decided to increase the score to 4

---

> > > ### Author Response · Authors · 2025-08-06
> > >
> > > We sincerely thank the reviewer for the positive comment and the improved rating.

---

### Official Review · Reviewer_5rXU · 2025-07-02

**Clarity:** 3
**Significance:** 3
**Originality:** 3
**Rating:** 5
**Confidence:** 4

**Summary:**

This paper proposes PPC, photography perspective composition, a model that can enhance the aesthetics of perspective in photography. The main idea is using a video generator to "warp" photographs with "suboptimal" perspective towards an aesthetically more pleasing composition, e.g., with the subject in the middle. The authors overcome the data scarcity problem for this approach by creatively "destructing" ground truth data (well-composed photographs, of which there are millions available) and then having the generator learn the inverse of that transformation. Overall, the paper is well written and executed, and the results look promising.

**Questions:**

see above

**Ethical Concerns:**

["NO or VERY MINOR ethics concerns only"]

**Final Justification:**

I remain positive on this paper.

The points that convince me:
- good novel use of video generators to create artifical, bad data for training the inverse task
- interesting problem statement, potentially with larger impoact to the field
- well grounded in technical literature, using eg RLHF, etc

**Limitations:**

yes

**Quality:**

3

**Strengths And Weaknesses:**

**Strengths**
- The idea is interesting and novel. I appreciate the authors branching out into the field of perspective composition by leveraging a video model, and thus being able to go beyond 2D-crop-based approaches. As the quality of video models will advance, so will the performance of this technique, which is why I think that this is an interesting and important paper.
- The paper leverages several existing techniques that have been proven to work well on different applications and in different contexts, e.g., RLHF, DPO, etc. As such, it is well grounded in the literature.
- I appreciate the detailed supplemental and the provided videos.

___

**Weaknesses**
- The current PQA method seems empirical, but will hopefully become less and less important as video generators keep improving in quality (see the first point under strengths).
- I feel that the fact that the model sometimes hallucinates / changes small details, such as the leaves in Fig. 7, can change the essence of the photograph slightly, which might be undesired for certain scenarios, e.g., portrait / wedding / pet photography. I'm wondering whether this will be a problem for close-ups of faces.
- In the supplemental video of the Taj Mahal, the PPC seems to overshoot the best perspective a bit. Have the authors experience similar issues, and is there a straightforward fix for this?

___

**Writing:**
- Please be careful with the word "optimal", especially when referring to sth as subjective as perspective (e.g., L. 13, 30, 33), since none of these are objectively, provably optimal.
- Starting a sentence with "While" usually implies a contradiction to follow, for example: While this seemed like a great idea initially, it later turned out to be wrong. Please adjust accordingly, see L. 21, 25, etc...
- A reference to cite for the related work section "Evaluation with Human Performance" is NICER: Aesthetic image enhancement with humans in the loop, by Fischer et al.

---

> ### Author Rebuttal · Authors · 2025-07-29
>
> We sincerely thank Reviewer 5rXU for recognizing the novelty of our work, particularly appreciating our innovative approach to photography composition through video models that transcend traditional 2D cropping methods. Below, we provide detailed responses to each concern. The "W" represents weakness.
>
> > W1. About the PQA method.
>
> We appreciate the reviewer for this point. We agree that as video generation models continue to advance, the reliance on PQA for quality assessment will naturally decrease. The improved capabilities of future video generation models will likely reduce the need for explicit quality filtering. Our technique will benefit from these continuous advancements in video generation models, making this research area promising for long-term development.
>
> However, we think that the evaluation model like PQA still has its unique value **beyond video generation models**.
> We are actively developing our next-generation system using **Unreal Engine 5-based** data collection pipelines to generate higher fidelity training scenes. In this context, **the evaluation model becomes particularly valuable for automated data collection workflows**: it can efficiently evaluate whether a generated scene is worth collecting by providing composition quality scores, enabling large-scale automated curation of high-quality training data without manual intervention. This automated scene assessment capability will be instrumental in scaling up our data collection efforts.
>
> > W2. About the model hallucination and detail changes affecting sensitive scenarios like portrait photography.
>
> We thank the reviewer for this concern about consistency issues. For sensitive scenarios like portrait photography,
> detail inconsistencies and hallucinations are inevitable.
> These issues primarily occur in background regions, especially when the model needs to generate background content not present in the original image. Actually, it is an inherent challenge for generative models.
>
> However, we find that these background detail distortions do not affect our understanding of the perspective transformation guidance provided by the model. This aligns with our core objective of providing perspective transformation guidance through videos.
>
> Additionally, to minimize the occurrence and impact of these consistency problems, we also make some efforts:
>
> **(1) Angle Limitation.** During data generation, we limit the maximum angle difference (20°) between the first and last frames of training videos. In our initial experiments without this limitation, the model often exhibited severe hallucinations by generating completely new scenes. By constraining the angle difference, we ensure the model learns to perform gradual perspective transformations rather than generating entirely different scenes, which significantly reduces hallucination issues. We found this angle limitation to be an effective solution for maintaining visual consistency while still allowing meaningful perspective changes.
>
> **(2) RLHF.** Even after implementing angle limitation, we still observe some videos with larger transformations that significantly affect the main subjects. Upon analyzing these problematic cases, we discover an interesting phenomenon: while these videos exhibit notable distortions, their perspective changes actually aligned well with human aesthetic preferences, but they deviated from the ground truth (GT) transformation directions. This led us to reconsider **whether strict GT adherence was necessary—perhaps alignment with human preferences would be sufficient**.
> Therefore, we introduce a two-stage RLHF training to relax the rigid GT constraints and allow the model to learn from human preference feedback rather than forcing exact GT compliance. This approach addresses the fundamental trade-off between GT fidelity and natural aesthetic transformations that humans find appealing.
> The effectiveness of this RLHF approach is demonstrated in Figure 6(a) in the paper: without RLHF, the cat's head directly deviates from its original position during perspective transformation, violating natural perspective relationships. After incorporating RLHF, the cat's head maintains proper perspective consistency, following realistic spatial transformations that align with human expectations.
>
> **(3) Future Improvements with Unreal Engine 5.** As mentioned above, we are developing our next-generation system using Unreal Engine 5-based data generation pipelines to further enhance scene consistency. This game engine approach provides significantly more controlled environments for generating training data. With perfect lighting control, we can eliminate the lighting inconsistencies that commonly occur in real-world 3D reconstruction. The precise camera tracking capabilities allow for exact trajectory control, ensuring consistent perspective transformations. Additionally, the engine maintains perfect structural consistency across all frames while avoiding the reconstruction artifacts and hallucinations that are inherent in current image-to-video approaches. This controlled data generation environment will be instrumental in reducing consistency issues.
>
> > W3. About the overshooting.
>
> We thank the reviewer for the careful review. Precise control over an exact "best perspective" is indeed challenging due to two main factors: (1) the subjective nature of perspective aesthetics itself, as different photographers may have varying preferences for what constitutes the "best" perspective for a given scene. (2) the inherent characteristics of video generation models.
>
> We initially explore a more straightforward angle prediction approach, designing classification models for four-directional angle predictions (5°, 10°, 15° increments). This classification-based method shows promising results and improved accuracy with increased data.
>
> **Angle Prediction Classification Results.**
>
> | Direction | Angle | 10K Data | 20K Data | 30K Data |
> |-----------|-------|----------|----------|----------|
> | Horizontal Left | 5° | 78.2% | 82.5% | 85.1% |
> | Horizontal Left | 10° | 74.6% | 79.3% | 82.7% |
> | Horizontal Left | 15° | 71.3% | 76.8% | 80.4% |
> | Horizontal Right | 5° | 77.9% | 81.8% | 84.6% |
> | Horizontal Right | 10° | 74.1% | 78.9% | 82.3% |
> | Horizontal Right | 15° | 70.8% | 76.2% | 79.9% |
> | Vertical Up | 5° | 76.4% | 80.7% | 83.8% |
> | Vertical Up | 10° | 72.8% | 77.6% | 81.2% |
> | Vertical Up | 15° | 69.5% | 74.9% | 78.6% |
> | Vertical Down | 5° | 75.8% | 80.1% | 83.3% |
> | Vertical Down | 10° | 72.2% | 77.1% | 80.8% |
> | Vertical Down | 15° | 68.9% | 74.3% | 78.1% |
>
> However, we find that simply predicting discrete angles do not align with our original intention. We aim to provide intuitive, educational guidance that demonstrates the transformation process, which led us to persist in adopting the video generation approach for creating instructional perspective transformation videos.
>
> **About the writing.**
>
> >1. Usage of the word "optimal" in subjective contexts.
>
> We appreciate the feedback on writing clarity and will replace "optimal" with "improved" or "enhanced perspective" throughout the paper to avoid subjective claims.
>
> >2. Logical issues with sentences starting with "While".
>
> We will revise the "While" problem and further polish our paper.
>
> For L21, the revised version is: "Master photographers, such as those in Magnum Photos, require professional knowledge and extensive training, making quality photography expensive and challenging for ordinary people."
>
> For L25, the revised version is: "Numerous approaches have been developed for image cropping, including saliency-based methods, learning-based techniques, and reinforcement learning strategies."
>
> >3. Missing NICER reference in the related work section.
>
> We will add the NICER reference (Fischer et al.) to the related work section on "Evaluation with Human Performance".

---

> > ### Author Response · Authors · 2025-08-06
> >
> > Dear Reviewer,
> >
> > Thank you once again for reviewing our paper. We would greatly appreciate it if you could take a moment to review our feedback and newly added experiments, and let us know if any concerns remain.
> >
> > Best regards,
> >
> > The Authors

---

> > > ### Comment · Reviewer_5rXU · 2025-08-06
> > >
> > > Hi, thanks for the clarifications, I appreciate the additional info on the direct angular prediction and agree with the authors that this would indeed be a much less desirable workflow than the proposed method. I remain positive on this paper.

---

> > > > ### Author Response · Authors · 2025-08-06
> > > >
> > > > We sincerely thank the reviewer for the positive comment and recognition of our work.

---

### Note · Authors · 2025-08-12

Dear ACs and Reviewers,

We sincerely thank the ACs and Reviewers for the careful review and constructive feedback on our paper.

In this paper, we propose photography perspective composition (PPC), a novel paradigm that extends beyond 2D cropping-based methods to achieve perspective adjustments. Cropping-based approaches are limited to operating within the image plane and fail when scenes contain poorly arranged subjects. Three key contributions are presented in our work: (1) An automated framework for building PPC datasets through expert photographs; (2) A video generation approach demonstrating transformation from suboptimal to better compositions; (3) A perspective quality assessment (PQA) model based on human performance.

The reviewers provided valuable comments and recognized the novelty of our work. The reviewers' concerns primarily focused on technical detail clarifications and implementation aspects. After our responses, the reviewers indicated that their main concerns were addressed.

We think our perspective-based composition opens up new possibilities for computational photography, and we hope it will inspire future research that makes professional composition skills more accessible to ordinary users.

Best regards,

The Authors

---

### Decision · Program_Chairs · 2025-09-17

**Decision:**

Accept (poster)

**Comment:**

Final rating: 5. Accept/ 4: Borderline Accept/ 4: Borderline Accept/ 4: Borderline Accept. The paper received initial ratings of  Accept/  Borderline Reject/ Borderline Accept/  Borderline Reject, which introduces a 3D recomposition method that adjusts viewpoint—rather than just cropping—to rebalance subject relationships while preserving true spatial layout. The paper addresses missing data and metrics by (1) auto-building PPC datasets from expert photos, (2) generating videos that show suboptimal→optimal perspective changes, and (3) training a human-grounded Perspective Quality Assessment model. The system works without prompts or camera trajectories, guiding everyday users to improve composition.

Reviewers agree the paper presents a novel and compelling idea (a view shared by all), supports it with extensive quantitative and qualitative experiments across diverse scenarios, and offers a well-designed pipeline. However, they find the validation thin—PQA appears empirical with no user studies or inter-annotator reliability—and several methodological details are under-specified (dataset size/splits, guidance integration, RLHF prompts, model identity). Key baselines are missing (state-of-the-art cropping-based and an image-only transformation), and practical fragility—artifacts, exposure/color shifts, hallucinations, perspective overshoot, and scale-related degradation—leaves real-world usability and generalizability uncertain.

After the rebuttal, all reviewers indicates their concerns were satisfactorily addressed and now lean toward acceptance, except reviewer rmgK, who still notes minor artifacts (e.g., structural distortion and loss of detail) in some visuals. The ACs  look into this and find that the authors proposed two mitigation strategies in the rebuttal, supported by additional quantitative evidence, which should reasonably improve visual quality.

Therefore, the area chairs agree to accept the paper and recommend the authors refine it according to the reviewers’ suggestions for the camera-ready version.